# Diversity and dissemination of viruses in pathogenic protozoa

Senne Heeren [1,2,3], Ilse Maes[1], Mandy Sanders[4], Lon-Fye Lye[5], Vanessa Adaui [6], Jorge Arevalo[7], Alejandro Llanos-Cuentas[7], Lineth Garcia[8], Philippe Lemey[2], Stephen M. Beverley [5], James A. Cotton [4,9], Jean-Claude Dujardin [1,3] ✉ & Frederik Van den Broeck [1,2] ✉

Viruses are the most abundant biological entities on Earth and play a significant role in the evolution of many organisms and ecosystems. In pathogenic protozoa, the presence of viruses has been linked to an increased risk of treatment failure and severe clinical outcome. Here, we studied the molecular epidemiology of the zoonotic disease cutaneous leishmaniasis in Peru and Bolivia through a joint evolutionary analysis of *Leishmania braziliensis* and their dsRNA *Leishmania* virus 1. We show that parasite populations circulate in tropical rainforests and are associated with single viral lineages that appear in low prevalence. In contrast, groups of hybrid parasites are geographically and ecologically more dispersed and associated with an increased prevalence, diversity and spread of viruses. Our results suggest that parasite gene flow and hybridization increased the frequency of parasite-virus symbioses, a process that may change the epidemiology of leishmaniasis in the region.

Viruses have the ability to infect any cellular life form on Earth. Particularly fascinating are RNA viruses that infect unicellular eukaryotes[1–3]. Some of these viruses have important biological roles, such as limiting fungal pathogenicity[4,5] or increasing protist fecundity[6]. The double-stranded RNA (dsRNA) viruses of the family *Totiviridae* have evolved extensive diversity and are present in phyla separated by a billion years of evolution, with closely related viruses identified in various genera of fungi and protozoa[7]. Totiviruses have no lytic infectious phase and thus adopted a lifestyle of coexistence favoring long-term persistence, passing from cell to cell mainly through mating and cell division[8]. Because of this intimate association, it has been postulated that these viruses have a mutualistic co-evolutionary history with their hosts[9].

Totiviruses encompass most viruses identified in pathogenic protozoa causing widespread severe illnesses such as trichomoniasis, giardiasis and leishmaniasis[7]. An iconic group of human pathogenic parasites is the genus *Leishmania* (family Trypanosomatidae), causing the vector-borne disease leishmaniasis in about 98 countries[10], mainly in the tropics and subtropics[11]. Members of the *Leishmania* genus are associated with the *Leishmania* RNA virus (LRV) (family *Totiviridae*). Despite reports of horizontal transmission[12–16], vertical transmission is thought to be the predominant mode of viral transmission resulting in general co-evolution of *Leishmania* and LRV[9,17,18]. Different types of LRV are carried by members of the *Leishmania* subgenera *Viannia* (LRV1), *Leishmania* (LRV2) and *Sauroleishmania* (LRV2)[16,19]. It was shown that the dsRNA of LRV1 is recognized by Toll-like receptor 3 (TLR3), which directly activates a hyperinflammatory response causing increased disease pathology, parasite numbers and immune response in murine models[20]. In human infections, the presence of LRV1 has been associated with an increased risk of drug-treatment failures and acute pathology[21–23]. The virus thus confers enhanced virulence and

[1]Department of Biomedical Sciences, Institute of Tropical Medicine, Antwerp, Belgium. [2]Department of Microbiology, Immunology and Transplantation, Rega Institute for Medical Research, Katholieke Universiteit Leuven, Leuven, Belgium. [3]Department of Biomedical Sciences, University of Antwerp, Antwerp, Belgium. [4]Welcome Sanger Institute, Hinxton, UK. [5]Department of Molecular Microbiology, Washington University School of Medicine, St. Louis, MO, USA. [6]Laboratory of Biomolecules, Faculty of Health Sciences, Universidad Peruana de Ciencias Aplicadas, Lima, Peru. [7]Instituto de Medicina Tropical Alexander von Humboldt, Universidad Peruana Cayetano Heredia, Lima, Peru. [8]Instituto de Investigación Biomédicas e Investigación Social, Universidad Mayor de San Simon, Cochabamba, Bolivia. [9]School of Biodiversity, One Health and Comparative Medicine, Wellcome Centre for Integrative Parasitology, College of Medical, Veterinary and Life Sciences, University of Glasgow, Glasgow, UK. ✉e-mail: jcdujardin@itg.be; fvandenbroeck@gmail.com

survival advantage to *Leishmania* in rodent models[24–26] and may play a key role in the severity of human leishmaniasis.

Given the epidemiological and biomedical relevance of the *Leishmania*-LRV1 symbiosis[27], there is a clear need to understand the diversity and dissemination of the virus in parasite populations. To this end, we jointly investigated the evolutionary history of *L. braziliensis* and LRV1 from Peru and Bolivia, using whole genome sequencing data. *Leishmania braziliensis* is a zoonotic pathogen circulating mainly in rodents and other wild mammals (e.g., marsupials) in Neotropical rainforests[28]. The parasite is the most prominent cause of cutaneous leishmaniasis in Central and South America and occasionally develops into the disfiguring mucocutaneous disease due to its spread to mucosal tissue. Our previous work in Peru has shown that LRV1 was present in >25% of the sampled parasites, and that the virus was significantly associated with an increased risk of treatment failure[22].

## Results

### Population genomics of *L. braziliensis* from Peru and Bolivia

We sequenced the genomes of 79 *L. braziliensis* isolates with known LRV1 infection status[22], which had been sampled during various studies on the genetics and epidemiology of leishmaniasis in Peru and Bolivia[29–33] (Supplementary Data 1, Supplementary Fig. 1). The read coverage ranged from 35x to 121x (median = 58x). Three isolates

(CUM68, LC2318, PER231) were removed because of aberrant alternate allele read depth frequencies (Supplementary Results). The resulting dataset (*N* = 76) comprised a total of 407,070 SNPs and 69,604 bi-allelic INDELs. The SNP allele frequency spectrum was dominated by low-frequency variants, with over 66% of SNPs being at <= 1% minor allele frequency. Chromosome and gene copy numbers were investigated using normalized median read depths. This revealed that the majority of chromosomes were disomic (Supplementary Fig. 2). When investigating variation in copy numbers for 8573 coding DNA sequences, we found 201–286 amplifications and 13–33 deletions per isolate (Supplementary Data 2). This is in line with what is known for the genomes of *Leishmania* spp[34].

A phylogenetic network based on genome-wide SNPs revealed a star-like topology whereby the majority of isolates were separated by long branches (Fig. 1a), a pattern symptomatic of recombination. Indeed, levels of linkage-disequilibrium (LD) were relatively low ($r^2$ decayed to <0.1 within <20 bp after correcting for sample size) (Supplementary Fig. 3a, b) and distributions of per-site inbreeding coefficients per population were unimodal and centered around zero (Supplementary Fig. 4a–d, Supplementary Data 3), after correcting for population structure and spatio-temporal Wahlund effects. Our genome-scale data thus indicate that the distinct populations in Peru and Bolivia are approximately in Hardy-Weinberg and linkage

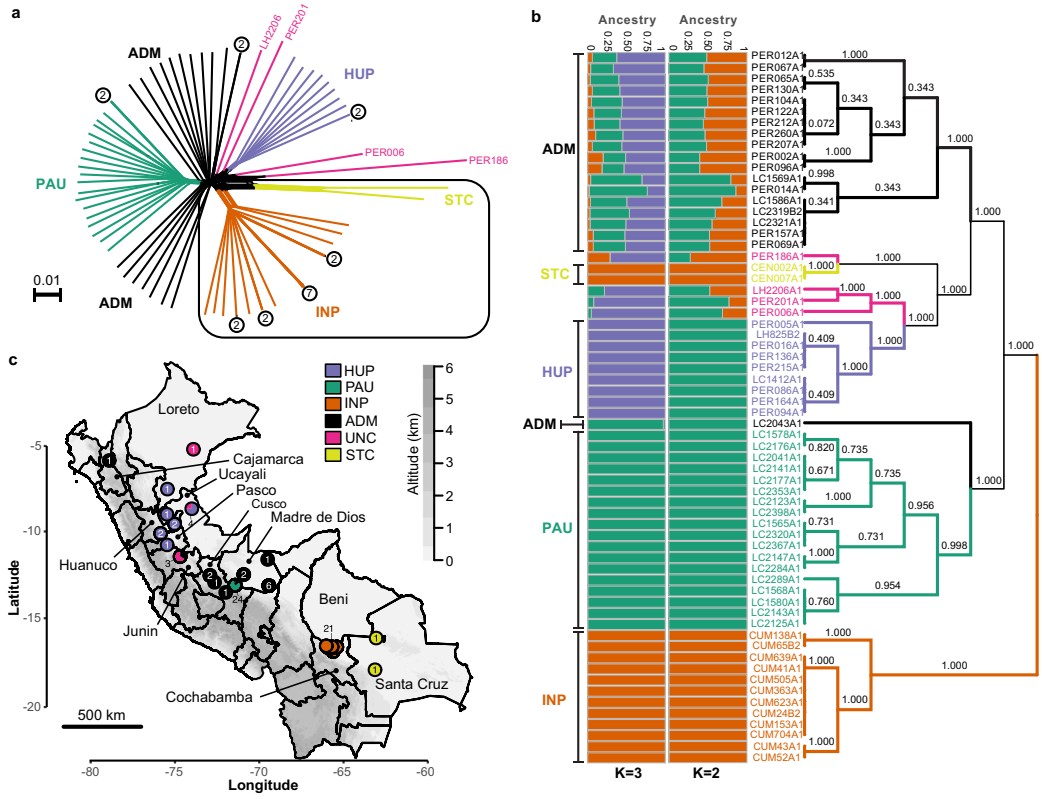

**Fig. 1 | Population genomic structure and admixture in *L. braziliensis* from Peru and Bolivia. a** Phylogenetic network as inferred with SPLITSTREE based on uncorrected *p*-distances between 76 *L. braziliensis* genomes (excluding isolates CUM68, LC2318 and PER231) typed at 407,070 bi-allelic SNPs. Branches are colored according to groups of parasites as inferred with ADMIXTURE, fineSTRUCTURE and PCAdmix (Fig. 2a). Box indicates the position of the Bolivian genomes; all other genomes were sampled in Peru. Circles at the tips of seven branches point to groups of near-identical genomes and show the number of isolates. The terminal branches are colored according to the parasite groups as shown in (**b**, **c**), while the colors of the internal branches were left black. **b** ADMIXTURE barplot summarizing the ancestry components assuming *K* = 2 or *K* = 3 populations in 65 genomes (i.e., excluding near-identical genomes). The phylogenetic tree summarizes the

fineSTRUCTURE clustering results based on the haplotype co-ancestry matrix (Supplementary Fig. 5). Numbers indicate the MCMC posterior probability of a given clade. Braces indicate the three ancestral populations (INP, HUP, PAU) and two groups of admixed parasites (ADM, STC); the remainder of the parasites were of uncertain ancestry (UNC). **c** Geographic map of Peru and Bolivia showing the origin of the 76 genomes. Dots are colored according to parasite groups as inferred with ADMIXTURE, fineSTRUCTURE and PCAdmix (Fig. 2a). Gray -scale represents altitude in kilometers, indicating the position of the Andes along the Peruvian and Bolivian Coast. Names are given for those Peruvian and Bolivian departments where a parasite was isolated. Country-level data for Peru and Bolivia, including administrative boundaries and altitude, were available from: http://www.diva-gis.org/Data. Source data are provided as a Source Data file.

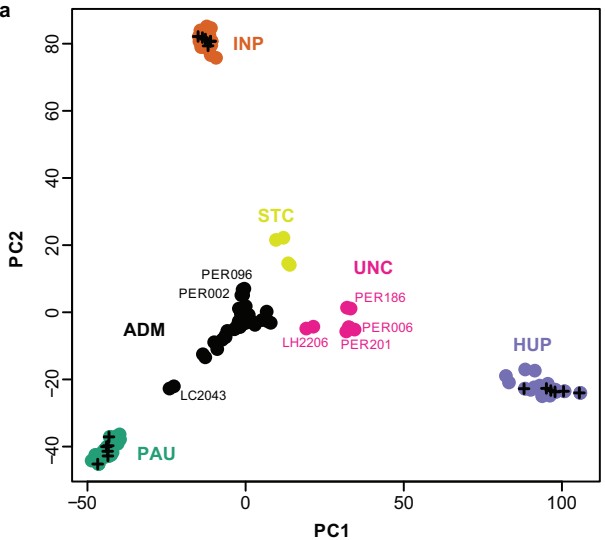

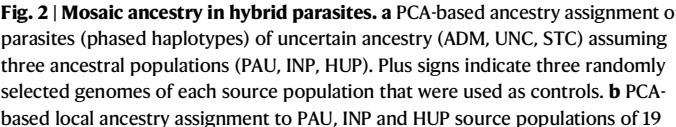

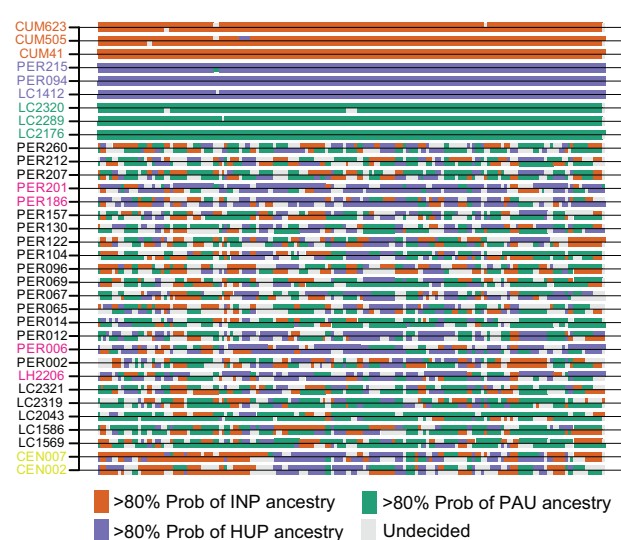

**Fig. 2 | Mosaic ancestry in hybrid parasites. a** PCA-based ancestry assignment of parasites (phased haplotypes) of uncertain ancestry (ADM, UNC, STC) assuming three ancestral populations (PAU, INP, HUP). Plus signs indicate three randomly selected genomes of each source population that were used as controls. **b** PCA-based local ancestry assignment to PAU, INP and HUP source populations of 19 ADM, 4 UNC, 2 STC isolates and three randomly selected isolates from each source population. Ancestry was assigned in windows of 30 SNPs along chromosome 35 (examples for other chromosomes are shown in Supplementary Fig. 7). Source data are provided as a Source Data file.

equilibrium, suggesting that recombination may be a prevalent and universal process in this species.

While the majority of *L. braziliensis* genomes differed at an average 9866 fixed SNP differences (median 9494), we also identified seven small groups of isolates with near-identical genomes that clustered terminally in the phylogenetic network (Fig. 1a, circles with numbers). Isolates within each of these groups displayed no or few fixed SNP differences and a relatively small amount of heterozygous SNP differences (Supplementary Data 4), and are thus likely the result of clonal propagation in the wild. Most of these clonal lineages were geographically confined (Fig. 1a, Supplementary Data 1) and one of them was sampled over a period of 12 years, suggesting that the observed clonal population structure is temporally stable (Supplementary Data 4).

We included one isolate from each clonal lineage and removed SNPs showing strong LD, leaving a total of 176,143 SNPs for investigating parasite population structure. Based on ADMIXTURE and fineSTRUCTURE analyses, we identified three distinct ancestry components (PAU, HUP, INP) (hereafter referred to as ancestral populations) corresponding to parasite groups with geographically restricted distributions (Fig. 1b, c, Supplementary Fig. 5). These three groups showed signatures of spatial and temporal genetic structure: PAU ($N = 19$) was sampled between 1991 and 1994 in Paucartambo (Southern Peru), INP ($N = 21$) was sampled between 1994 and 2002 in the Isiboro National Park (Central Bolivia) and HUP ($N = 10$) was sampled between 1990 and 2003 in Huanuco, Ucayali and Pasco (Central Peru) (Fig. 1b, c, Supplementary Fig. 5). Assuming $K = 2$ ancestry components, the two Peruvian groups PAU and HUP were clustered as one (Fig. 1b).

While our results indicate that *L. braziliensis* may be genetically clustered at sampling sites, we also identified three groups (ADM, UNC, STC) with mixed ancestries, hereafter referred to as hybrid groups to contrast with the identified ancestral populations (Fig. 1b): one large group of 20 isolates (ADM) sampled between 1991 and 2003 mainly from Southern Peru (Junin, Cusco and Madre de Dios), four isolates (UNC) sampled between 2001 and 2003 from Central/Northern Peru (Junin, Ucayali and Loreto) and one group of two isolates (STC) sampled in 1984/1985 from the Santa Cruz Department in Central Bolivia (Fig. 1c). The genetic diversity of the largest group of hybrid parasites (ADM) was much larger compared to the three ancestral populations:

the number of mitochondrial haplotypes and nuclear segregating sites was higher in the ADM group (309,543 SNPs; 5 haplotypes) compared to that of the PAU (189,748 SNPs; 4 haplotypes), INP (185,927 SNPs; 4 haplotypes) and HUP (215,741 SNPs; 2 haplotypes) populations. The observation of a large number of nuclear SNPs and mitochondrial haplotypes in the ADM group suggests that these parasites are descendants from multiple independent hybridization events (Supplementary Fig. 6).

We used PCAdmix to infer the genome-wide ancestry of the admixed isolates and three control isolates from each source population (Fig. 2a, b). Ancestry was assigned to phased haplotype blocks of 30 SNPs by comparing them to the reference panels of PAU, INP and HUP. While the control samples were assigned to their respective populations (92.9–99.4% of these haplotype blocks), the admixed individuals showed mixed ancestries between PAU (21.1–73.1%), INP (14.3–36.7%) and HUP (0.1–54.4%). Plots of local ancestry revealed a complex and heterogeneous pattern of mosaic ancestry between the three sources (Fig. 2b, Supplementary Fig. 7), suggesting that the hybrid parasites experienced many cycles of recombination following the initial hybridization event(s).

We next used the three-population statistic $f_3$ to formally test the potential source populations for introgressed alleles in the ADM and STC groups. When testing (test; A,B), a negative result indicates that the test group is an admixed population from A and B. We found a significantly negative $f_3$ statistic when ADM was the test group with PAU/HUP ($f_3 = -0.0013$, Z-score = −18.2) and PAU/INP ($f_3 = -0.0002$, Z-score = −2.8) as sources, but not with INP/HUP ($f_3 = 0.0013$, SD = , Z-score = 14.8) as sources. All estimated $f_3$ statistics were positive when STC was used as the test group. Hence, the ancestry of STC remains largely unresolved, but may involve admixture with divergent *L. braziliensis* lineages not captured in this study, as indicated by their distant position in a haplotype network of mitochondrial maxicircle SNPs (Supplementary Fig. 6) and by the comparatively large number of fixed SNP differences between STC and other parasite groups (Supplementary Fig. 8).

## Ancestral parasite populations are isolated in pockets of suitable habitat

The strong signatures of geographical isolation of the three inferred ancestral populations (PAU, INP and HUP) prompted us to elucidate

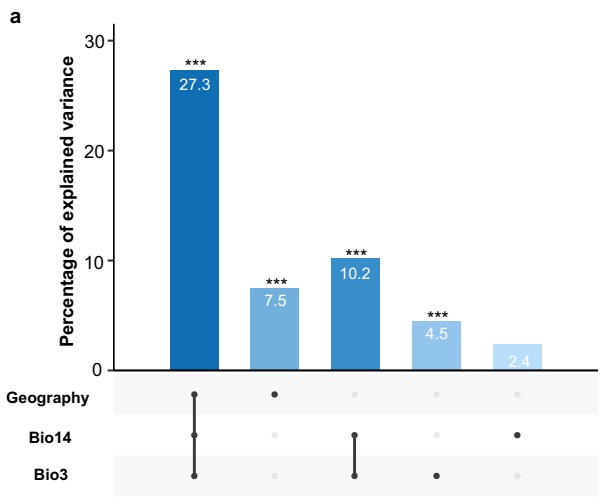
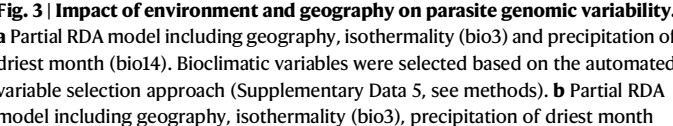
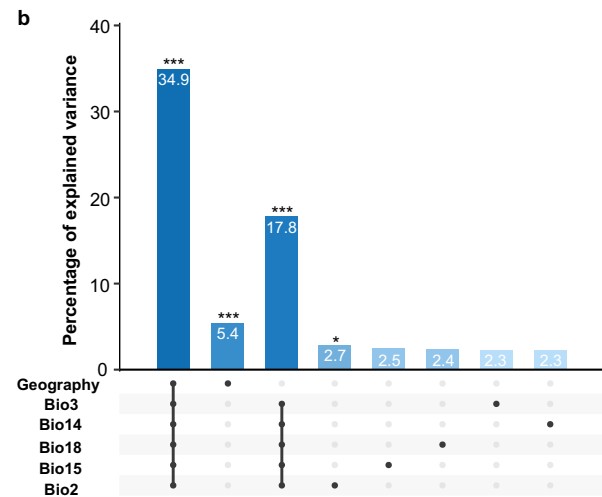

**Fig. 3 | Impact of environment and geography on parasite genomic variability.** **a** Partial RDA model including geography, isothermality (bio3) and precipitation of driest month (bio14). Bioclimatic variables were selected based on the automated variable selection approach (Supplementary Data 5, see methods). **b** Partial RDA model including geography, isothermality (bio3), precipitation of driest month (bio14), precipitation of warmest quarter (bio18), precipitation seasonality (bio15) and annual mean diurnal range (bio2). Bioclimatic variables were selected based on the manual variable selection approach (Supplementary Data 5, see methods). Source data are provided as a Source Data file.

the role of geography and the abiotic environment on the population genomic structure of *L. braziliensis* in the region. This was done through redundancy analysis (RDA) and generalized dissimilarity modeling (GDM), including geographic distance and 19 bioclimatic variables (Supplementary Data 5). Variable selection using two approaches consistently pinpointed differences in isothermality (bio3) and precipitation of driest month (bio14) between sampling locations as the main environmental contributors to parasite genetic distance. One of the two approaches also selected annual mean diurnal range (bio2), precipitation seasonality (bio15) and precipitation of the warmest quarter (bio18) (Supplementary Data 6; Supplementary Results). The RDA model including only bio3 and bio14 revealed that the environment and geography combined explained one-third (27.3%) of the genomic variability, of which 10.2% was contributed by environmental differences and 7.5% by geographic distance (Fig. 3a, Supplementary Data 7). The alternative RDA model, including bio3, bio14, bio2, bio15 and bio18, revealed a stronger environmental contribution in explaining the genomic variation. Here, the model could explain 34.9% of the total genomic variation where 17.8% was explained by the environmental component and 5.4% by geography (Fig. 3b Supplementary Data 7). It is notable that the partitioned contributions are not additive, indicating that there is a strong confounding effect among the environmental and geographic components in both models. This means that about one-third of the explainable genomic variation cannot be attributed to the individual components. Similar results were obtained with the GDM models (Supplementary Results).

In accordance with the RDA and GDM analyses, we generated environmental niche models (ENMs) using present and past bioclimatic variables to predict and spatially project the putative suitable habitat distribution of the ancestral populations at present and during the Last Glacial Maximum (LGM; 21 kya) and the Last Interglacial period (LIG; 130 kya) (Suppl Results). The present-day predicted suitability regions (Fig. 4a) coincided with tropical rainforests (Af), as predicted by the Köppen-Geiger (KG) climate classification (Fig. 4b), and shows that PAU, INP and HUP are surrounded by the less suitable tropical monsoon (Am) and savannah (Aw) forests and the non-suitable Andean ecoregions. Compared to the present, suitability predictions revealed a more widely distributed habitat for the LIG period and extreme retraction of habitat suitability during the LGM period

(Supplementary Fig. 9). This alternation of a widespread occurrence of suitable habitat during warmer climatic periods and a strong fragmentation in colder periods could have played a role in the divergence of *L. braziliensis* in the region.

We found that the three hybrid groups were ecologically and geographically more widespread compared to the ancestral parasite populations that were largely associated with tropical rainforests. The ADM group was mainly found within tropical rainforests (Af), tropical monsoon forests (Am), tropical savannah (Aw) and temperate (dry winter and warm summer) (Cwb) regions surrounding the location of the PAU population (Southern Peru). The UNC group is found within tropical rainforests (Af) and tropical monsoon forests (Am) surrounding the location of the HUP population (Central/Northern Peru). The STC group is found in tropical monsoon forests (Am) near the INP population (Central Bolivia). The geographic distances among sampled hybrids from the ADM (0–598 km) and UNC (0–694.1 km) groups were significantly larger compared to the distances among isolates from the PAU (0 km) and INP (0–66.4 km) groups, though not the HUP group (0–358.2 km) (Supplementary Data 8).

## LRV1 consists of divergent viral lineages that co-diverged with *Leishmania* host species

A total of 31 out of 79 analyzed *L. braziliensis* isolates (39%) were LRV1 +, as revealed in previous work[22]. While these originated mainly from Peru (*N* = 27), the geographic distribution of LRV1+ and LRV1- isolates overlapped in most localities (Supplementary Fig. 1). Here, we recovered LRV1 genomes for 29/31 LRV1+ isolates from Peru and Bolivia following a de novo assembly of dsRNA sequencing reads (Supplementary Data 9; Supplementary Results). The procedure failed for two isolates, either because of difficulties in growing cultures (PER096) or because the assembly yielded a partial LRV1 genome (PER231). Two different isolates (CUM65 and LC2321) each harbored two LRV1 genomes (Supplementary Results), differing at 999 (for CUM65) and 60 (for LC2321) nucleotides, bringing the total to 31 viral genomes. While only 0.004–0.1% of the RNA sequencing reads aligned against the LRV1 assemblies, median coverages were relatively high, ranging between 31X and 868X (median = 372X) (Supplementary Data 9). Sequences assembled from the RNA sequencing data were identical to ~1 kb sequences obtained with Sanger sequencing, confirming the high quality of our assemblies (Supplementary Results).

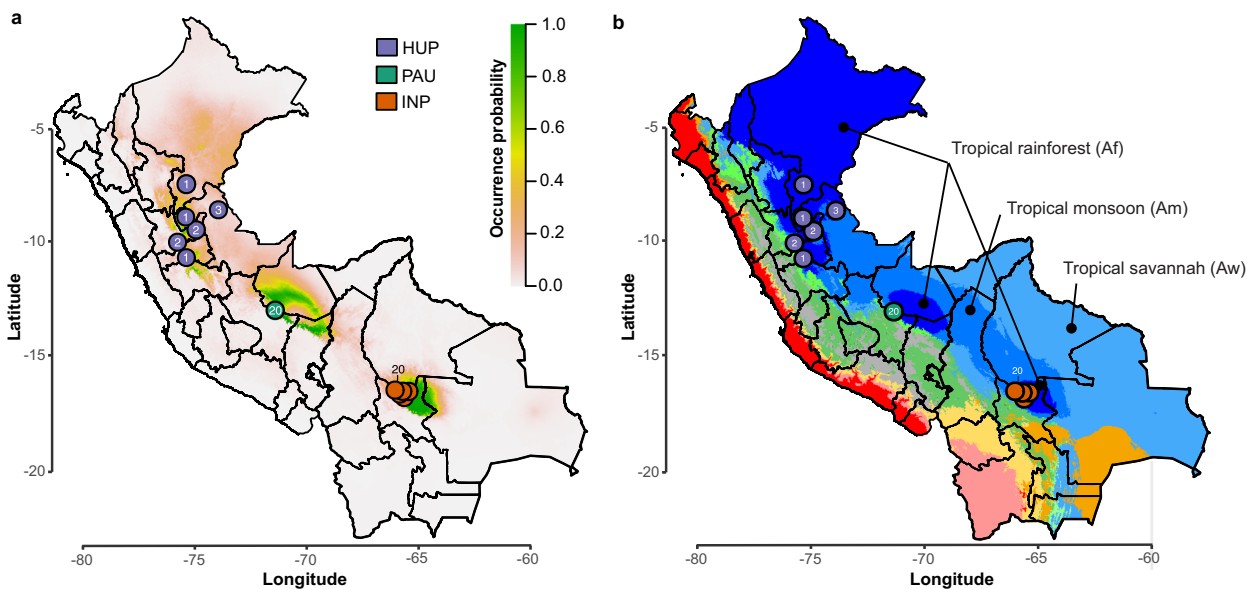

**Fig. 4 | Distribution of ancestral parasite populations in Peru and Bolivia.**
**a** Present-day predicted suitability regions (rm = 1.5) for *L. braziliensis* as revealed with ecological niche modeling. The continuous-scale legend represents habitat suitability (probability of occurrence). Map shows the distribution of isolates belonging to the three ancestral parasite populations HUP, PAU and INP. **b** Base map represents ecological regions in Peru and Bolivia as per Köppen-Geiger climate classification[115]. Only three ecological regions were labeled for visibility reasons. The distribution is shown for isolates (colored circles) belonging to the three ancestral parasite populations HUP, PAU and INP. Country-level data for Peru and Bolivia, including administrative boundaries, were available from: http://www.diva-gis.org/Data.

The assembled LRV1 sequences were 4738–5285 bp long, covering the near-full-length coding sequence of the virus, and showing an average GC content of 46% (45.4%–46.8%) (Supplementary Data 9). All but two genomes showed a typical totivirus organization, with two overlapping open reading frames encoding the Capsid Protein (CP; 2229 bp) and the RNA-dependent RNA Polymerase (RDRP; 2637 bp). Two isolates (PER014 and PER201) each contained 1 internal stop codon at amino acid positions 882 (TAG) and 875 (TAA), respectively. Both are located in the overlapping region of CP and RDRP within ORF3 (i.e., coding for RDRP). Sequence identities between these novel LRV1 genomes from Peru and Bolivia, and a previously published LRV1 genome from French Guiana (YA70; KY750610) ranged between 80% and 81%. Amino acid identity of both genes against YA70 ranged between 94% and 96% for CP, and 85%–87% for RDRP (Supplementary Data 9).

Initial viral evolutionary analyses were done using partial (N = 70) and full-length genome (N = 57) sequences, including publicly available LRV1 genomes from *L. braziliensis*, *L. guyanensis* and *L. shawi* from Brazil, French Guiana and Suriname. Despite the allopatric sample, phylogenies based on complete genomes (Fig. 5a) and partial sequences (Supplementary Fig. 10) revealed that LRV1 consists of divergent lineages that are grouped by *Leishmania* host species. In order to investigate whether LRV1 co-diverged with *L. braziliensis* and *L. guyanensis*, we reconstructed phylogenies based on viral genomes and their corresponding host genotypes. Similar to the results obtained for LRV1 (Fig. 5a), a phylogenetic network revealed a clear dichotomy between *L. braziliensis* and *L. guyanensis* (Supplementary Fig. 11). Moreover, co-phylogenetic analyses revealed a split of both viral and parasite genomes at the deepest phylogenetic node, confirming that LRV1 strains cluster with their *Leishmania* host species (Fig. 5b). A subsequent permutation test for co-speciation confirmed the topological similarity between both phylogenetic trees (RF = 70, p-value = 0.001). Similarly, we observed a significant but much weaker co-phylogenetic signal between LRV1 and *L. braziliensis* (RF = 38, p-value = 0.031). Closer inspection of the phylogenetic trees shows that the majority of incongruences

between LRV1 and *L. braziliensis* are linked to the hybrid ADM parasite group (Supplementary Fig. 12).

## Diversity and geographic spread of LRV1 in Peru and Bolivia

Focusing on LRV1 genome diversity from Peru and Bolivia, we defined a total of nine divergent viral lineages (L1-9), all supported by high bootstrap values (Fig. 6a) and low pairwise genetic distances (<0.09; Supplementary Data 10; Supplementary Data 11). No evidence of recombination was found in our set of LRV1 sequences, following pairwise homoplasy index (PHI) tests (p = 0.99)[35]. The number of LRV1 lineages per locality was positively correlated with the number of sampled parasites (Pearson's r = 0.76; t = 5.72; df = 24; p = 6.77e-06) (Supplementary Fig. 13). For instance, the most densely sampled location (Paucartambo, Cusco, Peru) in terms of *L. braziliensis* (N = 25) also contained the most viral lineages (Supplementary Fig. 13), two of which (L5 and L9) were only found in this location (Fig. 6b). Other locations that include multiple viral lineages are Tambopata (Madre de Dios, Peru) (L3 and L4) and Moleto (Cochabamba, Bolivia) (L7, L8) (Fig. 6b; Supplementary Fig. 13). These results show that multiple divergent viral lineages could co-occur within the same geographic region, and that a more dense sampling of parasites may uncover more viral lineages.

The majority of viral lineages (L2-L6, L8-L9) were found in a single locality (Fig. 6b), suggesting that the geographic spread of most LRV1 lineages is restricted. Two viral lineages were more widely distributed: one large group of viral genomes (L1) was found along the Andes from Northern Peru to Southern Peru, and one group (L7) was found in Cusco (Peru) and Cochabamba (Bolivia) (Fig. 6b). Three viral genomes of lineage L7 were virtually identical: viral sequences from the Bolivian isolates CUM68 and CUM65 were identical, and differed by one nucleotide from a viral sequence of the Peruvian LC2321 strain. The distal position of the two Bolivian L7 strains within a larger clade of Peruvian viral lineages (L3, L4, L5, L6) suggests that the former was introduced in Bolivia from Southern Peru (Fig. 6a, b). Finally, nearly all lineages (L2-L9) were found in the tropical rainforests (Af) of Peru and Bolivia, except for the widespread lineage L1 that was found in tropical

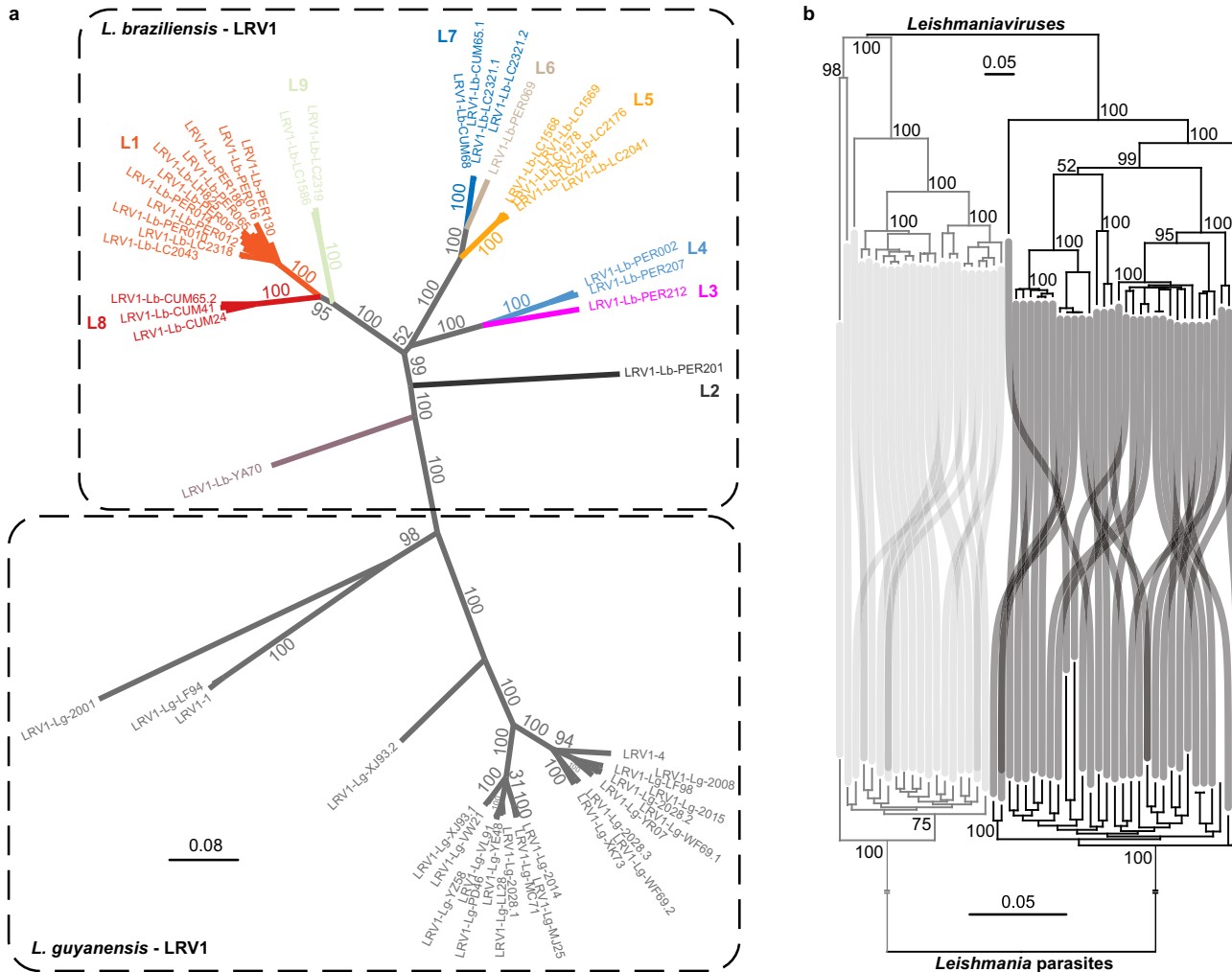

**Fig. 5 | Co-divergence of *Leishmania* and LRV1. a** Unrooted maximum likelihood phylogenetic tree based on 31 LRV1 genomes of *L. braziliensis* from Peru and Bolivia, one LRV1 genome (YA70) of *L. braziliensis* from French Guiana and 26 LRV1 genomes of *L. guyanensis* from Brazil, Suriname and French Guiana. LRV1 sequences of *L. braziliensis* are colored according to the different viral lineages identified in this study (L1-L9) (see Fig. 6). Branch values represent bootstrap support based on 100 replicates. **b** Phylogenetic tangle plot revealing clear patterns of virus-parasite co-divergence between *L. braziliensis* and *L. guyanensis* and their respective LRV1 clades. The gray and black branches and tree links correspond to *L. guyanensis* and *L. braziliensis*, respectively. Branch values represent bootstrap support based on 100 replicates.

rainforests (Af), tropical monsoon (Am), tropical savannah (Aw) and temperate climate (Cwb) (Fig. 7a). This suggests that LRV1 predominantly evolved in the lowland tropical rainforests before spreading to other ecological regions.

We reconstructed the viral dispersal dynamics in continuous space and estimated the impact of each lineage on the diffusion coefficient. This revealed that viral lineages L1 and L7 had the strongest influence on overall LRV1 dispersal in the region, exemplified by a tremendous drop in the diffusion coefficient when excluding L1 or L7 from the phylogeny (Fig. 7b, c; Supplementary Data 12).

**High LRV1 prevalence and lineage diversity within groups of hybrid parasites**

When investigating the distribution of viral lineages across the different *L. braziliensis* groups, we found two impactful observations. Firstly, the LRV1 prevalence significantly differed between the ancestral (PAU, HUP, INP) and hybrid (ADM, UNC) groups (Chi-squared test: $\chi^2 = 15.76$; df = 1; $p = 7.18e{-}05$). Specifically, we observed a significantly lower prevalence of LRV1 in the ancestral parasite populations PAU (26.3%; 5/19), INP (14.3%; 3/21) and HUP (20%; 2/10) compared to ADM (80%; 16/20), though not compared to UNC (50%; 2/4) due to the low sample size

(Fig. 8a; pairwise Fisher's exact tests; Supplementary Data 13). Secondly, the three ancestral populations PAU, INP and HUP (comprising a total of 50 isolates) were dominated by a specific viral lineage: the two LRV1+ isolates from HUP comprised the L1 viral lineage, the five LRV1+ isolates from PAU comprised the L5 viral lineage and the three LRV1+ isolates from INP comprised the L8 viral lineage, with the exception of isolate CUM65 that was coinfected with a viral genome from lineage L7 (Fig. 8a). Contrastingly, the LRV1 + ADM parasites harbored almost all viral lineages (L1, L3-L9), four of which were found exclusively in the ADM group (Fig. 8a). This is also reflected by a higher Shannon diversity index for the ADM group compared to the ancestral populations (Supplementary Data 13).

Finally, we examined the association between LRV1 and treatment outcome for all patients. A total of 41 isolates (51.8%) were sampled from patients with known treatment outcomes[22] (Supplementary Data 1). Of these, 23 patients were classified as cured while treatment failed for 18 patients. Similarly as described before[22], the percentage of treatment failures was lower for LRV1- isolates (37%; 9 failure vs. 15 cured) than for the LRV1+ isolates (53%; 9 failure vs. 8 cured), although this difference was not significant (exact logistic regression: $p = 0.36$) due to our lower sample size compared to the study of ref. 22. Here, we

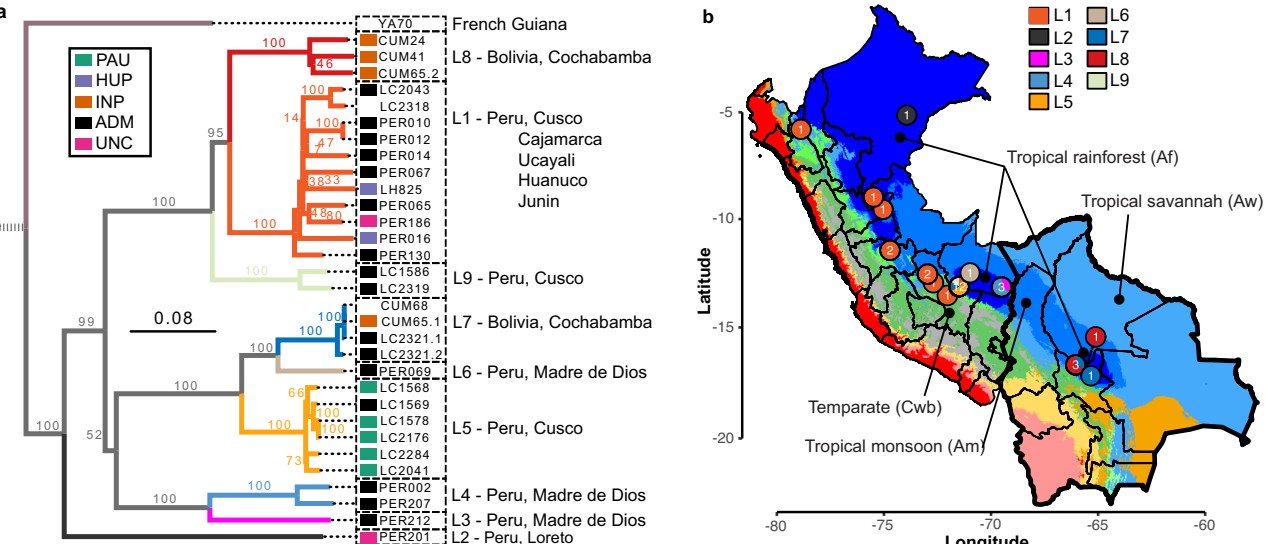

**Fig. 6 | Evolutionary history of LRV1 in Peru and Bolivia. a** Midpoint-rooted maximum likelihood phylogenetic tree based on 31 LRV1 genomes of *L. braziliensis* from Peru and Bolivia, one LRV1 genome (YA70) of *L. braziliensis* from French Guiana and 26 LRV1 genomes of *L. guyanensis* from Brazil, Suriname and French Guiana (the latter were omitted for visibility reasons). The position of the root is indicated with a dashed line. Clades are colored according to the different viral lineages (L1-L9). Branch values represent bootstrap support based on 100 replicates. The scale bar depicts the number of substitutions per site. Colored boxes at the tips of each branch represent the population structure of *L. braziliensis* (see

legend), with colors matching the different groups of parasites as shown in Fig. 1. Note that the tetraploid hybrid parasites LC2318 and CUM68 from Peru and Bolivia, and isolate YA70 from French Guiana were omitted from the analyses of parasite population structure; these were thus not assigned to any parasite group. **b** Base map represents the ecological regions in Peru and Bolivia as per Köppen-Geiger climate classification[115]. Only four ecological regions were labeled for visibility reasons. The distribution is shown for the nine viral lineages L1-L9 (colored circles). Country-level data for Peru and Bolivia, including administrative boundaries, were available from: http://www.diva-gis.org/Data.

investigated in more detail the distribution of treatment failures across the different LRV1 lineages and parasite groups. This was only done for INP (67% with known treatment outcomes), HUP (70%), ADM (70%) and UNC (100%) for which sufficient data on treatment outcome was available; PAU and STC were excluded here because treatment outcome was unknown for all patients. An exact logistic regression analysis revealed that the difference in the percentage of treatment failures between parasite groups was marginally significant ($p = 0.08$). Specifically, we found that the hybrid groups ADM (57.1%: 8 failures vs. 6 cured) and UNC (75.0%: 3 failures vs. 1 cured) were more frequently associated with treatment failures compared to HUP (28.6%: 2 failures vs. 5 cured) and INP (35.7%; 5 failures vs. 9 cured). Remarkably, most of the patients with treatment failures (88%; 7/8 failures) in the ADM group were associated with LRV1, in particular the L1 viral lineage (63%; 5/8), while treatment failure was irrespective of LRV1 presence for the ancestral populations (Fig. 8b, c). Altogether, our results show that the hybrid ADM group was frequently associated with treatment failures, the majority of which occurred in patients infected with LRV1-bearing *Leishmania*.

## Discussion

Our main goal was to understand the evolution and dissemination of viruses in an important group of human pathogenic parasites. To this end, we studied the ancestry of natural *L. braziliensis* and LRV1 from Peru and Bolivia based on whole genome sequence analysis.

We firstly investigated the population genomic diversity and structure of *L. braziliensis*. Genetic diversity studies have shown that populations in Peru[30,36], Colombia[37] and Brazil[38,39] are genetically heterogeneous and structured according to their geographical origin. In addition, divergent ecotypes were described in Peru[32,33] and Eastern Brazil[40], suggesting that the environment may play a key role in the *L. braziliensis* population structure. In Peru, our previous work revealed the existence of two Andean and one Amazonian lineage[33], the latter of which was infected with LRV1[21,22]. Here, we show that Amazonian *L. braziliensis* is further subdivided into distinct ancestral

populations that are isolated in patches of tropical rainforest, confirming that this species primarily evolved within the Amazonian rainforest. Each parasite population contained about half the total number of SNPs identified in our panel, indicating that we only captured a part of its genomic diversity in the region, and suggesting that *L. braziliensis* is genetically heterogeneous. Models of landscape genomics show that geographic distance and in particular environmental differences between sampling locations contributed to partitioning parasite diversity. In addition, ecological niche models revealed that the suitable habitat of *L. braziliensis* has changed over the past 150,000 y, including major contractions during LGM that may have promoted the diversification of this parasite in the region. Our observations indicate that the extremely diversified ecosystem of the Amazonian forest, including various host–vector communities[41], together with forestation changes over the past 150,000 years may have driven the large diversity and population substructure of *L. braziliensis*.

It has been postulated that protozoan parasites may have a predominantly clonal mode of reproduction in natural populations and that sexual recombination events are rare[42], although this theory has been the subject of intense debate for more than 30 years[42–44]. For *L. braziliensis*, studies using multilocus microsatellite profiles revealed contradictory results, including moderate degrees of inbreeding in natural populations from Peru and Bolivia[30,36] and significant levels of recombination in populations from the Brazilian Atlantic Coast[39]. Our genome-scale data indicate that the ancestral populations in Peru and Bolivia are approximately in Hardy-Weinberg and linkage equilibrium, suggesting that recombination may be a prevalent process in this species. In addition, we identified 25 isolates (32% of our panel) with signatures of mixed ancestry, and demonstrated that these are the result from multiple independent hybridization events. Against this background of recombination, we found six groups with two and one group with seven near-identical genomes that are the result of clonal propagation. These observations suggest that Amazonian *L. braziliensis* follows an epidemic/semi-clonal model of evolution[45,46], as

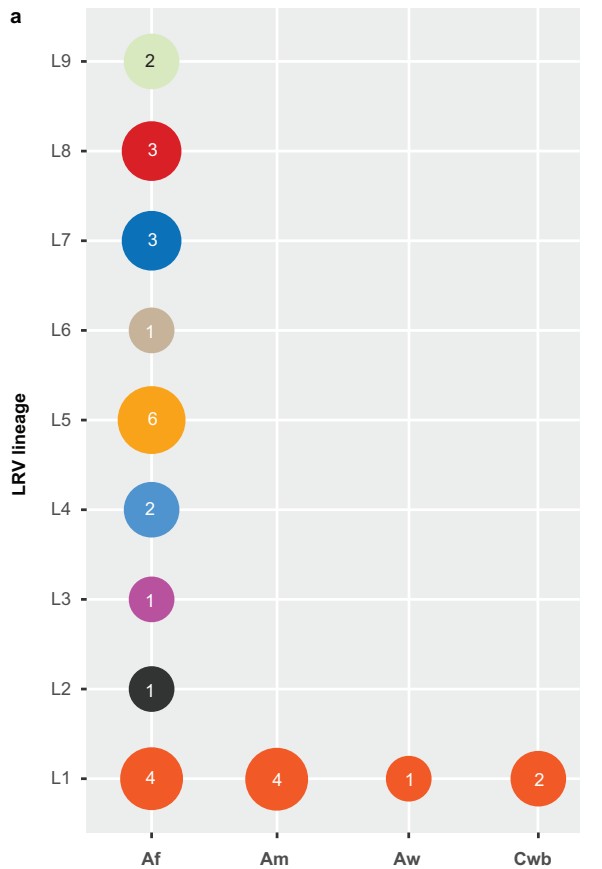

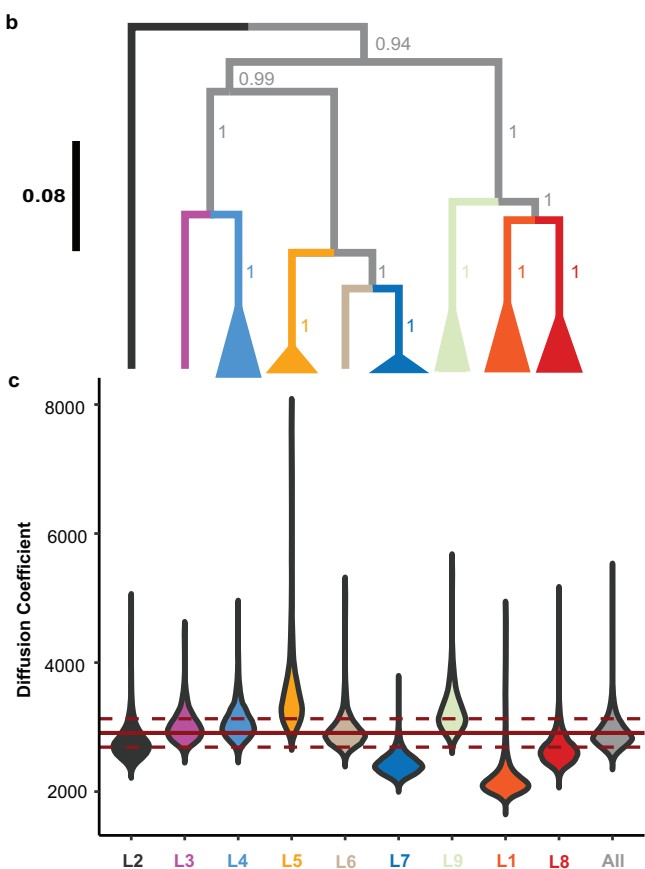

**Fig. 7 | Phylogeographic reconstruction of LRV1. a** LRV1 sample size structured per lineage and per ecological region as per Köppen-Geiger classification. Af tropical rainforest. Am Tropical monsoon forest. Aw Tropical savannah. Cwb Temperate, dry winter, warm summer. **b** The summarized maximum clade credibility (MCC) tree (excluding PER010) from the posterior tree distribution. Lineages are collapsed for visibility purposes and are color coded according to the x-axis in (**c**). Branch support values represent posterior probabilities. The black scale bar represents the number of substitutions per site. **c** Distribution of the inferred

diffusion coefficients of the whole LRV1 dataset (excluding PER010) and lineage-excluded partitions. The most right violin (gray) represents the diffusion coefficient distribution of the full LRV1 phylogeny. The other violins (colored per lineage) represent the change in the diffusion coefficient after removal of the specified lineage from the phylogeny. The solid and dashed red lines show the mean diffusion coefficient from the full LRV1 phylogeny and the standard deviation, respectively. Source data are provided as a Source Data file.

proposed for other protozoan parasites[44]. This model assumes frequent recombination within all members of a given population, where occasionally a successful individual increases in frequency to produce an epidemic clone.

We next investigated the diversity and evolution of LRV1. The presence of LRV1 has been demonstrated for several *L. Viannia* spp. such as *L. braziliensis* and *L. guyanensis*[21,22,47–51], as well as *L. panamensis*[52,53], *L. shawi*[51], *L. naiffi*[54] and *L. lainsoni*[51]. Phylogenetic analyses based on partial sequences from Brazil and French Guyana revealed that LRV1 mainly clustered according to their respective *L. (Viannia)* host species[55–57]. Here, we added sequence data from Peru and Bolivia and confirmed that LRV1 lineages cluster according to the different *L. (Viannia)* species, indicating that horizontal transfer of LRV1 between parasite species is rare[55,56]. The agreement between LRV1 and *Leishmania* phylogenies corroborates the general hypothesis that LRV1 may have co-evolved with *L. Viannia* spp[9,19,58]. While we observed a clear pattern of co-divergence at the parasite species level, we detected a much weaker intraspecific co-phylogenetic signal between LRV1 and *L. braziliensis*. Our data shows that the majority of sequenced LRV1 genomes and lineages were sampled from hybrid parasites, and that co-phylogenetic incongruences are mainly caused by prevalent horizontal transmission of LRV1 due to parasite gene flow and hybridization. Such intraspecific phylogenetic incongruences have also been observed for LRV2 in *L. major* from Uzbekistan[59] and other endosymbionts, such as *Wolbachia*[60].

Probably the most remarkable outcome of our study is that hybrid parasites were geographically and ecologically widely distributed and commonly infected with strains from a pool of genetically diverse LRV1, while ancestral populations were confined to the tropical rainforests and infrequently associated with a single LRV1 lineage. This suggests that (i) reproductively isolated parasite populations may lose viral diversity through genetic drift[61,62], a process that could explain the absence of LRV1 in *Leishmania* species that experienced population bottlenecks and reproduce mostly clonally, such as *L. peruviana*[33] and (ii) parasite gene flow and hybridization may maintain and/or replenish LRV1 diversity in the region. The latter is best exemplified by the most prevalent viral lineage L1 that is the only viral lineage found in ecological regions other than the tropical rainforests, and the L7 viral lineage that was introduced in Bolivia from Southern Peru. These two viral lineages also showed much higher dispersal rates compared to the others and are frequently associated with hybrid parasites. These results suggest that parasite gene flow and hybridization might have mediated the spread of the two viral lineages in new foci. One observation of potential concern is that 61% (11 of 18) of the patients experiencing treatment failure were infected with hybrid parasites, half of which (5 of 11) infected with the widespread L1 viral lineage. This indicates that patients may be more at risk of treatment failure when infected with hybrid *Leishmania* parasites that carry the L1 viral lineage, although our sample size is too low to statistically confirm this trend at present.

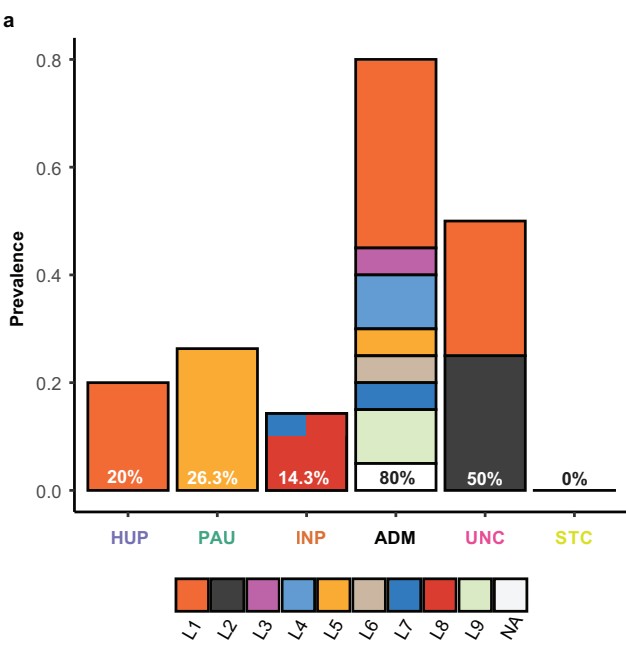

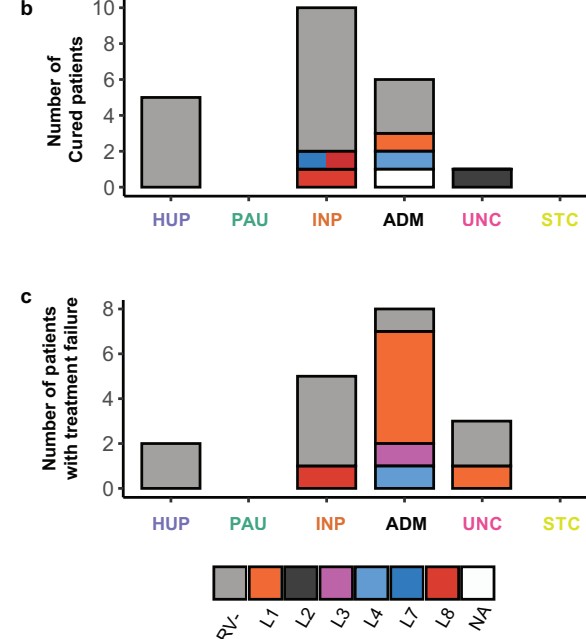

**Fig. 8 | Viral prevalence and lineage diversity among parasite groups.**
**a** Prevalence of LRV1 in each of the inferred parasite groups. Occurrence of clinical cure (**b**) and treatment failure (**c**) for the different parasite groups. In all panels, the colored stacked bars represent the contribution of each viral lineage. Source data are provided as a Source Data file.

We propose that environmental destruction and migration of humans and hemerophile reservoir hosts, such as dogs and rats[36,63], during the past century may have resulted in the displacement of *L. braziliensis* and their viruses out of the tropical rainforests. In the 19th century and earlier, the Peruvian Amazon region was sparsely populated and poorly integrated into the Peruvian nation[64]. Ever since the rubber boom in the 1870s, Peru has been characterized by migrations of inhabitants (for the purposes of farming, agriculture or mining) and military personnel (for the purpose of narcotic and guerilla control) to Amazonian departments, including migrations from Andean highlands to Amazon regions and also between the latter[64]. The activities of the migrants were often linked to an expansion of deforested areas[65], a known risk factor for vector-borne diseases. In addition, it was shown that recent labor migrants (arrival <5–6 years) working in the Peruvian Department of Madre de Dios (for e.g., gold mining, woodwork) were more susceptible to *Leishmania* infections compared to residents. This was mainly linked to the poorer living conditions of migrants (e.g., sleeping outdoors or in camps, lack of sanitation) compared to residents (e.g., better housing conditions, higher immunological protection)[66,67]. Human migrations may thus play an important role in the dissemination and secondary contacts of the zoonotic *Leishmania* parasites and their viruses throughout the region.

In conclusion, our results show that *L. braziliensis* hybrids are associated with an increased prevalence, diversity and spread of LRV1. This may have profound epidemiological consequences in the region because the presence of LRV1 has been linked to the severity of human leishmaniasis[23,25,26,51] and drug treatment failure[21,22,54]. Within a broader context, our work adds to a growing body of evidence indicating that parasite hybridization is a major public health concern[68].

## Methods
### Ethics statement
Samples were obtained from cutaneous and muco-cutaneous leishmaniasis patients who presented at health facilities for care. Sample collection was done during previous studies (IC 18.CT 96.0123 and ICA4-CT-2001-10076) on the genetics and epidemiology of leishmaniasis in Peru and Bolivia[29–33]. At the time, informed consent was obtained from all study participants. The study protocol was authorized by the ethical committees of Peruvian (Instituto de Medicina Tropical Alexander von Humboldt, Lima) and Bolivian (Centro Universitario de Medicina Tropical, Cochabamba) partners, and approved by the Institutional Review Board of the Antwerp Institute of Tropical Medicine.

### Statistics and reproducibility
No statistical method was used to predetermine sample size. No data were excluded from the analyses. The experiments were not randomized. The Investigators were not blinded to allocation during experiments and outcome assessment.

### Nucleic acid extraction and sequencing of DNA parasites and their dsRNA viruses
This study included 79 *L. braziliensis* isolates (Supplementary Data 1) from Peru ($N = 55$) and Bolivia ($N = 24$) that were sampled within the context of various studies on the genetics and epidemiology of leishmaniasis. In Bolivia, the majority of isolates ($N = 21$) were sampled between 1994 and 2002 within the context of a cutaneous leishmaniasis outbreak in the Isiboro National Park (Department of Cochabamba). Two isolates were sampled between 1984 and 1985 within the Santa Cruz Department and one is of unknown origin (Supplementary Fig. 1). In Peru, more than half of the isolates were sampled between 1991 and 2003 in the Cusco Department ($N = 29$), mainly from the Paucartambo province ($N = 25$). The remaining 26 isolates originated from Madre de Dios ($N = 9$), Ucayali ($N = 5$), Huanuco ($N = 4$), Junin ($N = 4$), Loreto ($N = 2$), Pasco ($N = 1$) and Cajamarca ($N = 1$) (Supplementary Fig. 1).

All 79 isolates were cultured for 3–4 days in the HOMEM medium (Gibco) with Fetal Bovine Serum (FBS; 20%) at the Antwerp Institute of Tropical Medicine. Parasites were subjected to a small number of passages (mean = 18 ± 5) to reduce potential culture-related biases in parasite genomic characterization[69]. Parasite cells (ca. $10^7$–$10^8$ parasites/ml) were centrifuged into dry pellets and their DNA was extracted using the QIAGEN QIAmp DNA Mini kit following the manufacturer's protocol. At the Wellcome Sanger Institute, genomic DNA was sheared

into 400–600 base pair fragments by focused ultrasonication (Covaris Inc.), and amplification-free Illumina libraries were prepared[70]. One hundred base pair paired-end reads were generated on the HiSeq 2000, and 150 bp paired end reads were generated on the HiSeq ×10, both according to the manufacturer's standard sequencing protocol.

Previous work has shown that 31 out of the 79 isolates were positive for LRV1[22]. The majority of these LRV1+ isolates originated from Peru ($N = 27$) while the remaining four were sampled in the Isiboro National Park (Cochabamba, Bolivia) (Supplementary Fig. 1). More than half of the Peruvian LRV1+ isolates originated from Cusco ($N = 15$) of which 11 were sampled in the Paucartambo province. LRV1 was also detected in isolates from Madre de Dios ($N = 5$), Junin ($N = 3$), Cajamarca ($N = 1$), Huanuco ($N = 1$), Loreto ($N = 1$) and Ucayali ($N = 1$) (Supplementary Fig. 1).

The 31 LRV1-positive isolates[22] were grown, as above, for 2–3 weeks (mean passage number $= 18 \pm 5$) at the Antwerp Institute of Tropical Medicine, ensuring high parasite yields (ca. $10^7$–$10^8$ parasites/ml). Isolation of dsRNA was performed as previously described[71]. In short, isolation involved a TRIZOL reagent (Invitrogen) RNA extraction followed by a RNase-free DNase I (NEB) and a S1 nuclease (Sigma-Aldrich) treatment along with an additional purification step (Zymoclean Gel RNA Recovery kit; Zymo Research). Double-stranded RNA of approximately 5.2 kb was visualized on 0.8% agarose gel (TAE buffer) stained with ethidium bromide. Extracts of dsRNA were sequenced at Genewiz (Leipzig, Germany) using the NovaSeq 6000 platform (Illumina) generating on average 35,665,319 150 bp paired end reads per isolate.

## Bioinformatics and population genomics of *L. braziliensis*

Sequencing reads were mapped against the *L. braziliensis* M2904 reference genome, as conducted earlier[33], using SMALT v.0.7.4 (https://www.sanger.ac.uk/tool/smalt-0/). The reference genome assembly comprises 35 chromosomes (32.73 Mb) and a complete mitochondrial maxicircle sequence (27.69 kb), and is available at https://tritrypdb.org/[33,72] as TriTrypDB-46_LbraziliensisMHOMBR75M2904_2019. Genome wide variant calling (SNPs, INDELs) was done using GATK v.4.0.1.0[73,74]. More specifically, we used GATK HaplotypeCaller for generating genotype VCF files (gVCF) for each isolate. Individual gVCF files were combined and jointly genotyped using CombineGVCFs and GenotypeGVCFs, respectively. SNPs and INDELs were separated using SelectVariants. Low quality variants were excluded using GATK VariantFiltration following GATK's best practices[75] and BCFtools v.1.10.2[76] view and filter. Specifically, SNPs were excluded when QD < 2, FS > 60.0, SOR > 3.0, MQ < 40.0, MQRankSum < −12.5, ReadPosRankSum < −8.0, QUAL < 250, format DP < 10, format GQ < 25, or when SNPs occurred in clusters (ClusterSize=3, clusterWindowSize = 10). INDELs were excluded when QD < 2, FS < 200.0, ReadPosRankSum < −20.0. The final sets of SNPs and INDELs were annotated using the M2904 annotation file with SNPEFF v4.5[77]. At heterozygous SNP sites, the frequencies of the alternate allele read depths[34] were calculated using the vcf2freq.py script (available at github.com/FreBio/mytools).

Chromosomal and gene copy number variation were estimated using normalized read depths. To this end, per-site read depths were calculated with SAMtools depth (-a option)[76]. Haploid copy numbers (HCN) were obtained for each chromosome by dividing the median chromosomal read depth by the median genome-wide read depth. Somy variation was then obtained by multiplying HCN by two (assuming diploidy). To obtain gene HCN, the median read depth per coding DNA sequence (CDS) was divided by the median genome-wide read depth. The HCN per CDS were summed up per orthologous gene group. Gene copy number variations were then defined where the z-score was lower than −1 (deletions) or larger than 1 (amplifications).

A NeighborNet network was reconstructed based on uncorrected *p*-distances (i.e., the proportion of sites where two sequences are different) with SplitsTree v.4.17.0[35]. The population genomic structure was examined with ADMIXTURE v.1.3.0[78] and fineSTRUCTURE v.4.1.1[79]. ADMIXTURE was run assuming 1 to 10 populations ($K$), performing a

5-fold cross-validation procedure, and after removing SNPs with strong LD as identified with plink v.1.9[80] (--indep-pairwise 50 10 0.3). The similar order of magnitude for the CV error values of $K = 1$–3 (Supplementary Data 14) prompted us to investigate the sub-structuring of *L. braziliensis* for $K = 2$ and $K = 3$. CHROMOPAINTER analysis (as part of fineSTRUCTURE) was run to infer the ancestry based on haplotype similarity. To this end, individual genotypes were phased with BEAGLE v.5.2[81] using default parameters, after which fineSTRUCTURE was run using 8 million MCMC iterations with 50% burn-in, and 2 million maximization steps for finding the best tree topology[82]. Local ancestry was assigned with PCAdmix[83] using phased genotype data (i.e., haplotypes) as obtained with the BEAGLE v5.4[81]. *F3*-statistics were calculated with Treemix v1.13[84]. Finally, Hardy-Weinberg equilibrium (HWE) was assessed by calculating the per-site inbreeding coefficient as Fis = 1 − Ho/He; with Ho representing the observed heterozygosity and He the expected heterozygosity. Decay of LD was calculated and visualized using PopLDdecay[85]. To control for spatio-temporal Wahlund effects, we calculated Fis and LD decay using subsets of isolates that were sampled close in time (year of isolation) and space (locality), and taking into account population genomic structure.

To investigate the spatio-environmental impact on genetic variation among the three ancestral parasite populations we calculated geographic distances among sampling locations and extracted 19 bioclimatic variables of the WorldClim2 database[86]. We firstly investigated the role of geography on the genomic differentiation of the ancestral components through linear and non-linear regression analyses of distance matrices (Supplementary Methods). Secondly, we used RDA[87] and GDM[88] to test the impact of environmental differences and geographic distance on parasite genetic distance. To account for model overfitting and multicollinearity, we performed two variable selection approaches (mod-A, mod-M) (Supplementary Methods). ENM was done based on partially jittered parasite occurrence data (excl. hybrid parasites), to account for the spatial uncertainty of isolates with non-unique coordinate pairs, along with present-day and past bioclimatic variables using Maxent v.3.4.3, as implemented in the 'dismo' R package[89,90]. A more detailed description on the landscape genomic analyses (variable selection, RDA, GDM and ENM) is presented in the Supplementary method section. Finally, we compared the geographic ranges between parasite groups by means of a Kruskal-Wallis test ('stats' R package[91]) followed by pairwise Dunn's tests ('FSA' R package[92]) with Benjamini-Hochberg (BH) corrected p-values.

## Bioinformatics and phylogenetic analyses of LRV1

Raw RNA sequencing reads were trimmed and filtered with fastp[93] using the following settings: a minimum base quality (-q) of 30; the percentage of unqualified bases (-u) set to 10; per read sliding window trimming based on mean quality scores (-5, front to tail; -3, tail to front) with a window size (-W) of 1 and mean quality score (-M) of 30; right-cutting reads (--cut_right) per 10 bp windows (--cut_right_window_size) when mean quality score (--cut_right_mean_quality) is below 30; only considering reads between 100 (-l) and 150 bp (-b). LRV1 sequences were assembled de novo with MEGAHIT[94] and identified using BLASTn[95] against conventional LRV1 reference genomes LRV1-1[96] and LRV1-4[97] (accession numbers M92355 and U01899, respectively). In order to check and improve the quality of the assemblies, trimmed reads were mapped against the LRV1 contigs with SMALT as described above, with the minimal nucleotide identity (-y) set to 95%. Alignments were examined with Artemis[98] and used to improve assemblies with Pilon v.1.23[99]. Genome sequences were aligned using the L-INS-i algorithm in MAFFT v.7.49[100].

For comparative purposes, we included 26 (near-) complete LRV1 genomes and 13 partial LRV1 sequences of *L. braziliensis* ($N = 8$), *L. guyanensis* ($N = 26$) and *L. shawi* ($N = 1$) from French Guiana, Brazil and Suriname[55,56,96,97]. Multiple sequence alignments were generated by trimming to the minimum sequence lengths and re-aligning (i) all

available genomes to 5189 bp sequences ($N = 57$) and (ii) all available sequences to 755 bp sequences ($N = 70$). Maximum likelihood (ML) trees were generated using IQ-TREE v.1.6.12[101] and 100 bootstrap replicates. Based on the lowest Bayesian Information Criterion, the ModelFinder module[102] revealed GTR + F + R4 as the best substitution model for the genome alignment. The GTR + F + R4 substitution model was also applied on the partial sequence alignment for consistency. Pairwise genetic distances among LRV1 genomes were calculated with the 'ape' R-package[103] (model = 'raw'). Viral genomes with a genetic distance below 0.09 (i.e., 9% of the sites that are dissimilar among two genomes) were grouped into distinct viral lineages. Nucleotide diversity and Fst statistics were calculated within and between viral lineages using the 'PopGenome' package in R[104] while recombination was tested by PHI tests implemented in SplitsTree[35,105].

Viral prevalences between parasite groups were compared through a Fisher's exact test ('stats' R package[91]) followed by pairwise Fisher's exact tests ('fmsb' R package[106]) with BH corrected p-values. We also compared the viral prevalence between the combined sets of the genetically distinct (PAU, HUP, INP) and hybrid (ADM,UNC) groups through a chi-squared test ('stats' R package[91]). The viral lineage diversity between parasite groups was described by species richness and the Shannon diversity index ('vegan' R package[87]). Finally, we investigated the extent of treatment failure associated with LRV1 through an exact logistic regression ('elrm' R package[107]) with 10,000 MCMC iterations (2000 burn-in).

## Viral phylogeographic inference in continuous space

To reconstruct the dispersal history of LRV1, we performed a Bayesian phylogeographic analysis in a continuous spatial framework (i.e., based on coordinate data) using BEAST v.1.10.4[108] and the BEAGLE library v.4.0.0[109,110], following a previously published protocol[111]. Due to a lack of a temporal signal in the data, sampling dates were set to zero (default setting). The nucleotide substitution model was set to the GTR substitution model with the base frequencies set to be empirically calculated and the site heterogeneity model to Gamma with four discrete categories. The trait evolutionary model (i.e., substitution model for location data) was set to the lognormal Relaxed Random Walk model where the bivariate trait represents longitude and latitude. Additionally, a random jitter (factor = 0.01) was added to the tips to add a slight noise to the data to avoid poor model performance, which is recommended when not all sampling points have unique coordinate pairs[111]. Next, the uncorrelated relaxed clock model[112] was selected and for the location partition all ancestral states were reconstructed. All priors and operator parameters were retained at their default values. Markov chain Monte Carlo sampling was run over ten million states, sampled every 1000 states, with a 10% burn-in. Subsequently, the posterior tree distribution was summarized using BEAST's TreeAnnotator constructing a maximum clade credibility tree.

BEAST was run ten times: one for the entire LRV1 phylogeny followed by a stepwise exclusion of each viral clade in order to compare the inferred diffusion coefficients between the different partitions. The differences in diffusion coefficient between the different phylogeny partitions were tested by means of a Kruskal-Wallis test followed by pairwise Dunn's tests with BH corrected p-values.

## Co-phylogenetic analysis of LRV1 and *Leishmania*

Co-phylogenetic analyses were done at both the species (between LRV1 infecting *L. braziliensis* and *L. guyanensis*) and at the population (LRV1 infecting *L. braziliensis* from Peru and Bolivia) level. These analyses were performed using the phytools R package[113]. To assess the co-evolutionary history of LRV1 with both parasite species, we complemented our dataset with 24 LRV1 genomes and 19 *L. guyanensis* and 1 *L. braziliensis* SNP genotypes from French Guiana (N = 19) and Brazil (N = 1)[55] (*L. guyanensis* reads accession: SRA - PRJNA371487; LRV1 genome accessions: GenBank - KY750607 to KY750630 [https://www.

ncbi.nlm.nih.gov/nuccore/KY750607,KY750608,KY750609, KY750610,KY750611,KY750612,KY750613,KY750614,KY750615, KY750616,KY750617,KY750618,KY750619,KY750620,KY750621, KY750622,KY750623,KY750624,KY750625,KY750626,KY750627, KY750628,KY750629,KY750630]). The phylogenetic trees were tested for topological similarity calculating the Robinson-Foulds distance[114], using the phytools package[113], between both trees with comparison against a null distribution of 1000 permuted un-correlated trees. For LRV1 we reconstructed a ML tree as described above, including the 23 LRV1 genomes of *L. guyanensis* and 32 LRV1 genomes of *L. braziliensis*. For *Leishmania*, sequence reads of *L. guyanensis* were mapped against the M2904 reference genome and GATK Haplotypecaller was run as described above. We then performed joint genotyping on a dataset including the 19 *L. guyanensis* genomes and 80 *L. braziliensis* genomes. SNPs were filtered following similar criteria as described above (QD < 2, FS > 200.0, SOR > 3.0, MQ < 40.0, MQRankSum < −12.5, ReadPosRankSum < −8.0, QUAL < 250, info DP < 10, ClusterSize = 3 and ClusterWindowSize = 10). The *Leishmania* ML tree was generated using IQ-TREE based on 7571 jointly called bi-allelic SNPs with 100 bootstrap replicates and GTR + F + R5 as best performing substitution model[101,102]. For the co-phylogenetic reconciliation at the intraspecific level, we focused on phylogenies generated for our dataset of *L. braziliensis* and LRV1 from Peru and Bolivia. Specifically, we used the GTR + F + R4 ML tree generated with IQ-TREE for LRV1 and the bifurcating tree generated by fineSTRUCTURE for *L. braziliensis*.

## Reporting summary

Further information on research design is available in the Nature Portfolio Reporting Summary linked to this article.

## Data availability

The metadata used for this study was the collection site for the 79 *L. braziliensis* isolates, which is given for each sample in Supplementary Data 1 (provided as Source Data file). Genomic sequence reads of the 79 sequenced *L. braziliensis* genomes are available in the European Nucleotide Archive (https://www.ebi.ac.uk/ena/browser/home) under accession number PRJEB4442. The assembled sequences of the 31 LRV1 genomes are available in GenBank (https://www.ncbi.nlm.nih.gov/genbank/) under accession numbers OQ673070-OQ673100 [https://www.ncbi.nlm.nih.gov/nuccore/OQ673070,OQ673071,OQ673072, OQ673073,OQ673074,OQ673075,OQ673076,OQ673077,OQ673078, OQ673079,OQ673080,OQ673081,OQ673082,OQ673083,OQ673084, OQ673085,OQ673086,OQ673087,OQ673088,OQ673089,OQ673090, OQ673091,OQ673092,OQ673093,OQ673094,OQ673095,OQ673096, OQ673097,OQ673098,OQ673099,OQ673100]. Source data are provided with this paper.

## Code availability

Key analyses scripts and input data for the landscape genomic analyses are available from https://github.com/sheerenbiol/LandGenLeish.

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

## Acknowledgements

This work received financial support from the European Commission (Contracts TS2-CT90-0315 and TS3-CT92-0129) and Directie-Generaal Ontwikkelingssamenwerking en Humanitaire Hulp (DGD) (Belgian cooperation). F.V.D.B. and S.H. acknowledge support from the Research Foundation Flanders (Grants 1226120 N and G092921N). L.F.L. and S.M.B. acknowledge support from the National Institute of Health (5R01AI130222 and 2R01AI029646). This research was funded in part by the Wellcome Trust Grant [206194]. For the purpose of Open Access, the author has applied a CC BY public copyright license to any Author Accepted Manuscript version arising from this submission.

## Author contributions

Substantial contributions to the conception and design of the work were made by F.V.D.B., J.C.D., S.M.B. and J.A.C. Substantial contributions to the acquisition of data were made by V.A., J.A., A.L.C. and L.G. Parasite culturing and nucleotide extractions were done by I.M. and S.H. Sequencing of parasite genomes was done by M.S. Sequencing of partial LRV1 sequences was done by L.F.L. Substantial contributions to the analysis and interpretation of the data were made by S.H. and F.V.D.B. Substantial contributions to drafting or revising the manuscript were made by S.H., F.V.D.B., J.C.D., P.L., V.A., J.C. and S.M.B.

## Competing interests

The authors declare no competing interests.
