## [Peer Review File · Nature Communications]

Diversity and dissemination of viruses in pathogenic protozoaREVIEWER COMMENTS

Reviewer #1 (Remarks to the Author):

In the manuscript entitled "Parasite hybridization promotes spreading of endosymbiotic viruses", Heeren et. al. report a genetic diversity analysis of *L. braziliensis* isolates from Brazil and Peru, and the association of Leishmania genetic diversity with LRV-1 diversity and content. This study contributes a substantial resource of sequence data for the Leishmania research community, which indeed will allow further analysis on population structure and association with specific phenotypes of interest in humans. Understanding the mechanisms of LRV-Leishmania co-evolution are indeed interesting and potentially relevant in the context of disease management for some *L. Viannia* species. Results from this study confirm previous observations showing that LRV diversity is associated with the geographical distribution of Leishmania subpopulations – and therefore Leishmania genetic diversity (as acknowledged by the authors). This study provides a thorough genetic analysis of Leishmania and LRV. From my point of view, it provides incremental understanding of Leishmania-LRV associations, but does not report a groundbreaking finding regarding the mechanisms of co-evolution the parasite and the virus, nor its relationships with human disease.

Based on the title of the paper (Parasite hybridization promotes spreading of endosymbiotic viruses), the data presented in this manuscript does not demonstrate a causal relationship between Leishmania hybridization and spread of LRV. It shows an association between Leishmania subpopulations and LRV lineages, which has been previously reported, though to a limited scale. It would have been interesting to see experimental data evaluating the permissiveness of specific Leishmania subpopulations to infection by the range of LRV lineages identified in this study, to further demonstrate specificity of the interactions and/or co-evolution or potential for enhanced "spreading".

In the abstract, authors state: "Our results suggest that parasite hybridization, likely due to recent human migration and ecological perturbations, increased the frequency of endosymbiotic interactions known to play a key role in disease severity." It is not clear how the data support this asseveration. There is no available data to show the relationship between the outcome of human infection and the Leishmania subpopulation/LRV-lineage interaction that could be associated with disease severity. Has it been demonstrated that "increased frequency of endosymbiotic interactions" plays a role in disease severity? A retrospective analysis of cases in the different geographical regions where strains were collected could serve to evaluate this. Is there higher frequency of disease severity in the areas where there is more LRV, more diversity of LRV or a specific Leishmania/LRV pair? Is there a difference in disease manifestation in places with presence of "clonal" vs. "hybrid" populations?

Reviewer #2 (Remarks to the Author):

The manuscript by Heeren et al is an interesting study of the genetic structure of *Leishmania braziliensis* in Peru and Bolivia coupled with the analysis of (*Leishmania* RNA virus 1) LRV1 distribution. The broad range of applied methods and the decent amount of analyzed isolates undoubtedly deserve credit. However, the manuscript has a number of important flaws and needs a thorough revision.

1) There is a wrong positioning of the manuscript and disbalance of the content. In general, it appears that there is one study nested within another. In principle, there could be two different manuscripts. According to the title, the study should be focused on the increased prevalence of the virus in hybrids. However, most of the voluminous text is devoted to the genetic structure of *L. braziliensis*. Moreover, the main conclusion of the manuscript that has been put to the title is not even properly presented, The authors provide just the percentages of LRV1-bearing isolates for different genetic groups of the studied parasite and claim that there are significant differences without applying any statistical analysis for that. I am embarrassed by the necessity to provide an

advice to researchers with an extensive expertise in statistics. However, this important message looks very weak. I believe that at least a simple Chi-squared test for proportions, as proposed by Campbell, 2007 is needed for the comparison of the two main groups: unmixed vs hybrid isolates. If the authors believe that their main finding concerns this fact from the title, a reader would expect a more extensive discussion of this phenomenon, which is actually absent. The abovementioned disbalance can be diminished by decreasing the amount of text in the "populational" part by removing unnecessary parts related to methods and/or supplementary results. For example, the very first section "Natural genome diversity of *Lb* parasites" is mostly redundant. I would also suggest to think about reformulating the title to mention there that the study includes the analysis of the populational structure of *L. braziliensis*.

2) As mentioned above the whole text is too voluminous even though a proper discussion is absent. Moreover, many of the results mentioned in the text are illustrated by only supplementary figures. The authors should think, whether these results are really not important and then, maybe move them to supplementary Results (see above), or show some of the figures in the main text. Of note, the journal allows up to 10 figures, but only 5 are included into the main text. In addition to what is suggested above, the authors should also reorganize the text in such a way that description of the existing *L. braziliensis* groups is not interrupted. In fact, the groups are first mentioned in the sentence starting on the line 187. The authors allegedly identified all of them by the ADMIXTURE/fineSTRUCTURE analysis. However, on the Fig. 1B UNC is not labelled (and apparently does not form a solid group) and the isolate LC2043A1 is not classified (ADM group?). Apparently, the definitive separation of the six discussed group became available only in the results of the PCAdmix analysis mentioned on the line 238. Why they are separated by other analyses remains obscure. This creates a big mess in a reader's head.

3) The section concerning "co-divergence" between *Leishmania* spp. and LRV1 was apparently added to the manuscript to somehow compensate the abovementioned disbalance in the analyses of *Leishmania* and LRVs. However, it looks redundant, since it is not logically connected with the other parts of the manuscript and, what is even more important, does not bring any new knowledge. The cited paper by Cantanhêde et al., 2018 has been already devoted to this issue and the new analysis despite the inclusion of additional sequences did not show anything new. The subdivision of LRV1 into lineages according to species has been demonstrated in the cited paper. Of note, this is only partially true, since the group of LRV1s from *L. guyanensis* show a very deep divergence and is non-monophyletic due to an isolate from *L. shawi*, which is nested within. In any case the analysis in Cantanhêde et al., 2018 is more comprehensive, since it considers not only to species of *Leishmania*, but also to geography.

4) The section named as discussion does not actually discuss the data, but repeats some of the statements from Results, which, in turn, do not represent pure results, but a mixture of results and discussion seasoned with a bit of methods. In a revised manuscript a proper discussion will be needed, which will consider also some results, which, for some reason were not of much use. I meant the inference of paleolandscapes, which should be considered for the explanation of ancient contacts between the three maternal populations of *L. braziliensis* described in the manuscript. It appears that the authors had something like this in mind, but then this was omitted. Of note, the claiming that "suitable habitats have been relatively stable over the past 130,000 years" does not correlate with the obtained results. The Fig. S10 shows that in the past the suitable areas were larger and that the present-day "islands" of optimal environmental conditions previously had contacts.

Now some medium-to minor concerns (the list is not comprehensive!):

1) The language of the manuscript needs improvement. There are some mistakes (e.g. "where" instead of "when"(line 80); missing articles; "on the European Nucleotide archive" and "on Genbank" (should be "in", lines 581 and 583); etc). In addition, the language is stylistically poor, with the same words used in one sentence up to three times. The abbreviation *Lb* for *Leishmania braziliensis* is a "mauvais ton" and must be avoided in the text. Although it may appear that the authors use it to shorten the sentences, in fact it is not true, since usually this abbreviation appears in a collocation "*Lb* parasites", which is approximately of the same length as "*L. braziliensis*", but reads very awkward. In most cases it is even not needed to mention the species, since the overwhelming part of the text is related only to this single species. Therefore, it is not

needed to specify, whose genome, groups, populations, etc. are meant. There are cases of incorrect usage of terms, repetitive sentences, etc. See some additional examples are given below.

- 2) The genetic groups discussed in the text must be referred as "groups", when listed, not as "populations" plus "groups". I understand, that the three of the groups represent maternal populations, by hybridization of which the other three appeared. This must be properly explained and the terms should be used carefully.
- 3) Line 51: totiviruses > Totiviridae first mentioning should be with the Latin name).
- 4) Line 53: yeasts are also fungi
- 5) Line 53: "closely related viruses identified in almost all genera of yeasts, fungi and protozoa" Seriously? In almost all genera of fungi and protozoa???
- 6) Line 60: Family > family
- 7) Line 64: " virus was most likely present in the common ancestor of Leishmania" – this is an obsolete view. The majority of Leishmania spp. do not have viruses. Some of those that have LRV2 (rare strains *L. tropica* and *L. infantum*) acquired it recently (the sequences are identical!) by lateral transfer from sympatric *L. major*. The situation with LRV1 also suggests at least such event between *L. guyanensis* and *L. shawi*. In addition, there are LRVs from monoxenous *Blechnomonas* spp. Thus, there were several lateral transfers in the evolution of this viral genus, therefore there is no reason to assume its inheritance from a common ancestor.
- 8) Lines 75 and 82: these sentences are virtually the same.
- 9) Line 95 (although this must not be in the main Results): specify that this concerns not all samples, but the majority of them.
- 10) Lines 115, 117: chromosomes cannot be diploid, this word refers to the state of the whole genome
- 11) Line 118: revise the sentence. The genomes were not entirely diploid if one of the chromosomes had a copy number other than double.
- 12) Line 123: " Variation in chromosome 123 and gene copy numbers was not associated..." – No data supporting this statement are provided. Either remove the sentence, or confirm it.
- 13) When you refer to figures, always mention a particular panel.
- 14) Fig. 1A: in general, it is virtually useless, since does not show much. It can be either removed or transferred to Supplementary figures. In any case it is unclear why geography is shown only for some isolates and, moreover, why for most cases departments are indicated (Cusco, Cajamarca, etc), but for one there is the name of province (Huancapallac). The colouring of the branches is weird. Why black is used not only for ADM, but also for some internal branches.
- 15) Fig. 1B must be larger (full-size) to make the isolate names readable at the print size. I would also suggest to mark groups not by rectangles, but by colouring branches using the same palette as on the left. This is especially important given that it is difficult to delineate there the UNC group and label the LC2043A1 isolate.
- 16) Fig. 1C. is also rather small, consider enlarging it. In addition, the circle with 24 isolates appears to have a pink component. If this is not a mistake, then as judged by this panel this one isolate from Cusco, plus two from Junin, one from Ucayali and one from Loreto sum up to five total UNC isolates, not four as mentioned in the text. Correct this discordance.
- 17) For the convenience of readers, provide the classification of isolates into the six groups in the Table S1.
- 18) Lines 128-129 (legend): the information is false, since not all groups could be inferred using ADMIXTURE and fineSTRUCTURE. Only subsequent PCA-based analyses provided this classification.
- 19) Line 132: "inferred at $K = 2$ or $K = 3$ populations": replace with "assuming 2 or 3 populations" for simplicity. Also simplify the text on the line 191. There is no need to use the K letter here, especially given that in ecology it may have different meanings.
- 20) Line 143: "It is clear" to whom? That is a bad style of writing.
- 21) Lines 144-147: The essence of the debate is not explained. The text reads like recombination is one of the modes of reproduction. Of note, this paragraph and the next one belong either to introduction or to discussion. Alternatively, the authors can decide, that they have a joint Results and Discussion section.
- 22) Lines 173-175: ungrammatical sentence.
- 23) Lines 178-179: ecotypes are entities occupying distinct ecological niches, so there is kind of tautology in this sentence.
- 24) Line 184: the abbreviation has been already explained
- 25) Line 185 and elsewhere: Present instead of Past tense (identify > identified).
- 26) Lines 191-192: "population" used thrice in a single sentence!

- 27) Line 195: why only for ADM details are provided?
- 28) Lines 196-215 (two paragraphs) must be moved elsewhere to separate this analysis from those focused on the genetic groups.
- 29) Lines 203-206: repeated colocations in one sentence. Revise it and somehow explain to the reader, that the values are not additive, since 27,3% includes also the combined effect of the two factors.
- 30) Line 209: revise the sentence. This analysis cannot say what habitats are suitable for a given species. It just predicts the spatial distribution of these habitats. So consider revising as "predicted putative suitable areas".
- 31) Line 210: LIG – unexplained abbreviation
- 32) Line 210-213: sentence is ungrammatical and with repeated words
- 33) Line 213: it is unclear whether savannah is considered as an unsuitable habitat (for some reason not even labelled on the map). However, STC isolates were obtained from tropical savannah.
- 34) In general, the text in this paragraph (Lines 208-215) must better explain what exactly the analysis is for.
- 35) Line 217 (legend): explain that the maps show only maternal populations (carefully use the terms), but not hybrids.
- 36) Fig. 2. In principle, it is unclear, why hybrids are not shown. Is not their distribution important? I do not think so. Moreover, I believe that the reconstructions of paleolandscapes from Figure S9 must be also shown here, at least with one of the rm values, although there should be no problem with space here. And provide their description in the text.
- 37) Line 226: populations vs groups (see above).
- 38) Lines 235-237: this conclusion is not justified. Not all of these haplotypes could be inherited. Apparently III and IX were inherited, while X, XII, and XIV-XVII could evolve afterwards.
- 39) Lines 238-239: individuals > isolates
- 40) Line 241: 92.9-99.4% of what?
- 41) Line 257: revise the sentence to make it clearer
- 42) Lines 262-264: Why median distances? Given the uneven sampling density they are more than confusing. Apparently, the distances are needed to claim that " hybrid parasites are ... more widespread". Then it is better to provide the maximal distance, or a range of distances between sampling locations, where a particular group was detected.
- 43) Lines 265-267: Why the authors do not consider contacts between the maternal populations in the past, when the suitable habitats could be connected to each other?
- 44) Fig. 3A: explain why the isolate LC2043 was classified as ADM. It looks isolated here and on the Fig. 1B. Also explain why the number of data points does not correspond to the number of isolates. For example. UNC should have four (or five? (see above)) isolates, but on this graph there are 7 points for this group.
- 45) Line 276: within what group of hybrid parasites? There are three of them
- 46) Line 281: dsRNA, not total
- 47) Most of the text in the first paragraph on the page 10 is redundant. These are mostly technical details, that could be transferred to methods or supplementary results.
- 48) Line 285: "0.04% (0.004%-0.1%)" > "0.004-0.1%"
- 49) Line 290: "covering the full-length coding sequence" – false, some of them (the shortest ones) contain only partial CDS
- 50) Line 303: the procedure/software for the homoplasmy index calculation is not mentioned in Methods
- 51) Line 304: "t=5.72" - What is this value?
- 52) Line 304: twice "sampled" in the sentence
- 53) Lines 305-311: discussion of the number of lineages for such minute geographical groups, many of which contain a single LRV+ isolate is not much useful.
- 54) Line 318: " The distal position of the two Bolivian L7 strains within a clade of Peruvian viral lineages" replace simply with "This"
- 55) Line 320: " all early-diverging lineages (L2-L9)" > "all lineages but L1". All lineages cannot be early-diverging.
- 56) Line 320: "was introduced" Apparently "must be "recently introduced"
- 57) Line 323: "temperate climate" – This is not labelled on the map. IN addition, it is not possible to see on this figure, where the isolates (or locations) belong, because the areas are tiny as compared to the size of the circles.

- 58) Line 331: "50 Lb isolates". Must be 51 (10+20+21)
- 59) " These observations strongly suggest" – No, this is not true. They cannot suggest this (moreover, strongly), since the CUM65 isolate (definitely from a single population) contains two different lineages! I would understand if you claimed that in each of these populations one viral lineage dominates (do not forget about an insufficient sampling).
- 60) Line 337: "contained almost all viral lineages (L1, L3-L9)" > "all viral lineages except for L2"
- 61) Line 339: gene flow is irrelevant here
- 62) Figure 4: numbers at branches are not explained
- 63) Line 358: repeating the unjustified statement from the introduction (see above)
- 64) Line 360: why "relatively" if there are identical sequences? Specify, that this concerns LRV2 (No more comments on this section, which should be removed)
- 65) Line 400: "two species" are not specified
- 66) Line 402: highly heterogeneous as compared to what? How do you know, where LRV1 predominantly evolved? It is better to mention, where you detected it, but this will not be the exhaustive list of locations, especially given that this virus occurs in other countries and other species of Viannia.
- 67) Lines 420-422: it is not possible to judge from the obtained data whether the hybridization was recent or ancient. The only conclusion that can be made at the current moment is that none of the hybrids was F1.
- 68) Line 441: 29 is not a half of 55. Specify the exact number.
- 69) Line 446: concentration of FBS? Any antibiotics supplemented?
- 70) Lines 453-455: awkward sentence with repeats
- 71) Line 460: LRV1 was not sampled, it was detected in some isolates
- 72) Line 462: do not repeat the cultivation conditions
- 73) Line 449 and 464: the number of cells collected?
- 74) Line 465 (reference 58): why this reference, which does not provide any additional details, but refers to an earlier paper?
- 75) Line 467: the name of the Zymoclean kit?
- 76) Line 470: number of reads generated?
- 77) Line 473-475: revise the sentence
- 78) Line 475: "the reference genome" > "the reference genome assembly
- 79) Line 477: specify the ID of the assembly (there can be more than one)
- 80) Line 486: "The final set... were" – ungrammatical
- 81) Line 491: reference for SAMtools
- 82) Line 499: " A NeighborNet tree" > "A Neighbor-Net network" or " A Neighbor-Net graph"
- 83) Line 501: why the results are shown only for K=2 and K=3?
- 84) Line 502: double "with". The sentence seems to be unfinished.
- 85) Line 504: "genetic ancestry" > "ancestry" (what else it can be, spiritual?)
- 86) Line 505: delete"computationally"
- 87) Line 507: "for finding ... for building" – revise
- 88) Line 510: examined > assessed; delete "the"
- 89) Line 514: "isolates ... isolated" – tautology; "year of isolation" > "year"
- 90) Line 517: included into what?
- 91) Line 542: specify the algorithm
- 92) Line 544: more inappropriate abbreviations of Leishmania spp. names...
- 93) "to 5189 bp ... and ... to 755 bp" – revise the sentence for clarity. Are these minimal lengths of sequences?
- 94) Line 550: Lines 547-551: revise for conciseness
- 95) Line 552: distanceS
- 96) Line 563: added to what?
- 97) Line 564: the accession number is wrong, also specify the database (GenBank)
- 98) Line 566 – used software is not specified
- 99) Lines 576-578: the sentence is vague
- 100) Data availability: submit the assembled maxicircle sequences and provide the accession numbers
- 101) Figure S11: why UNC is absent from the network?
- 102) Figure S14, legend, line 45: numbers are on the left, not on the right and they show not the count of positive L. braziliensis isolates, but the numbers of LRV1 lineages (as judged by the text of the manuscript). This was not easy to find out...

103) Figure S15. No legend for colours. Why French Guiana is coloured? What are the supports (do not appear as bootstrap percentages)?

Reviewer #3 (Remarks to the Author):

This study by Heeren and collaborators analyzed the hybridization of *Leishmania v. braziliensis* (Lb) protozoan parasite favoring the propagation of the endosymbiotic viruses (LRV1). This an important epidemiological study for investigators toward the comprehension on how LRV1 could play an important role in the pathological development of MCL and the appearance of atypic phenotype of the disease. Use of ecological biotope modelisation could contribute to the monitoring of geographic distribution of *Leishmania v.* and LRV1.

The manuscript is very well written, concise, and clear, and the results are presented in a structured and organized manner. The introduction provides a clear and informative context regarding *Leishmania* and the endosymbiont virus LRV1.

Comment and Suggestions

About the model,

An essential step for ecological niches modelisation consists of a statistical analysis toward the identification of all covariations of bioclimatic variables to select an optimal group of 19 variables necessary for the modelisation.

In their study, the authors did not mention this crucial step. This needs clarification.

Add the graphs regarding the most elevated gain for environmental variables in the model for each Lb population and explain in results how they influence the model.

Why did the authors not use the data about missing species?

Modelisation of niches should be based on a combination of presence or absence of species permitting a better determination of environmental preferences.

Explain the number of presence monitoring that they used for the modelisation.

As per, "van Proosdij AS, Sosef MS, Wieringa JJ, Raes N. Minimum required number of specimen records to develop accurate species distribution models. *Ecography*. 2016 Jun;39(6):542-52 ».

What they expect for appropriate habitat or potential distributions of Lb population regarding future bioclimatic variables?

Does different *Lutzomia* sps could explain some of the distribution obtained?

Why not all *viannia* are found with LRV1 endovirus? To which extend, in comparison to *Lviannia* without LRV1, the MCL development is different? *Lviannia* without LRV1 will also in a very similar proportion lead to MCL upon initial cutaneous lesion... This could be further discussed.

Rebuttal - LRV paper

Reviewer #1

In the manuscript entitled “Parasite hybridization promotes spreading of endosymbiotic viruses”, Heeren et. al. report a genetic diversity analysis of *L. braziliensis* isolates from Brazil and Peru, and the association of *Leishmania* genetic diversity with LRV-1 diversity and content. This study contributes a substantial resource of sequence data for the *Leishmania* research community, which indeed will allow further analysis on population structure and association with specific phenotypes of interest in humans. Understanding the mechanisms of LRV-*Leishmania* co-evolution are indeed interesting and potentially relevant in the context of disease management for some *L. Viannia* species. Results from this study confirm previous observations showing that LRV diversity is associated with the geographical distribution of *Leishmania* subpopulations – and therefore *Leishmania* genetic diversity (as acknowledged by the authors). This study provides a thorough genetic analysis of *Leishmania* and LRV. From my point of view, it provides incremental understanding of *Leishmania*-LRV associations, but does not report a groundbreaking finding regarding the mechanisms of co-evolution the parasite and the virus, nor its relationships with human disease.

We appreciate the reviewer’s acknowledgement of our thorough analysis and recognition of our incremental contribution towards understanding Leishmania-LRV associations. As outlined in the introduction and the first sentence of the discussion, the main goal of our study was to investigate the diversity and dissemination of LRV1 in Leishmania parasite populations. To the best of our knowledge, this is the first study examining the evolution of LRV1 intraspecifically at this scale (79 isolates collected across 25 localities in two countries). Previous studies focused mainly on the clustering of LRV1 interspecifically (i.e. across different Leishmania species) ¹⁻⁵, were conducted with limited numbers of isolates ^{1,5-8} or examined Leishmania and LRV sequence data independently ^{8,9}. We want to clarify that it was not our objective to investigate the mechanism of coevolution or its relationships with human disease, as stated by the reviewer, although we understand the origin of this confusion (as explained below).

Based on the title of the paper (Parasite hybridization promotes spreading of endosymbiotic viruses), the data presented in this manuscript does not demonstrate a causal relationship between *Leishmania* hybridization and spread of LRV. It shows an association between *Leishmania* subpopulations and LRV lineages, which has been previously reported, though to a limited scale. It would have been interesting to see experimental data evaluating the permissiveness of specific *Leishmania* subpopulations to infection by the range of LRV lineages

identified in this study, to further demonstrate specificity of the interactions and/or co-evolution or potential for enhanced “spreading”.

We agree with the reviewer that the title of the paper focuses only on one of the outcomes of our study and that we did not sufficiently address the relationship between Leishmania hybridization and spread of LRV1. As this concern was also raised by reviewer 2, we suggest a new title that better captures the breadth of our work: "Diversity and dissemination of endosymbiotic viruses in zoonotic Leishmania parasites". In addition, we provide additional statistical and Bayesian phylogeographic analyses to strengthen the link between parasite hybridization and LRV1 prevalence, diversity and spread (please see our response to the first comment of reviewer 2 where we provided more details about these analyses).

The reviewer suggests an experimental evaluation of the permissiveness of different Leishmania subpopulations to the LRV1 viral lineages identified in our study. We acknowledge that this would be interesting to investigate, and that our genomics work generated new hypotheses to be tested within an experimental framework. However, such work is extensive and should thus be part of a separate follow-up study. In the meantime, our work represents a first incremental step towards understanding the population dynamics of Lb and LRV1 in the wild.

In the abstract, authors state: “Our results suggest that parasite hybridization, likely due to recent human migration and ecological perturbations, increased the frequency of endosymbiotic interactions known to play a key role in disease severity.” It is not clear how the data support this asseveration. There is no available data to show the relationship between the outcome of human infection and the Leishmania subpopulation/LRV-lineage interaction that could be associated with disease severity. Has it been demonstrated that “increased frequency of endosymbiotic interactions” plays a role in disease severity? A retrospective analysis of cases in the different geographical regions where strains were collected could serve to evaluate this. Is there higher frequency of disease severity in the areas where there is more LRV, more diversity of LRV or a specific Leishmania/LRV pair? Is there a difference in disease manifestation in places with presence of “clonal” vs. “hybrid” populations?

We apologize for causing confusion about the relationship between disease outcome and Leishmania/LRV interaction. It was not our intention to claim that we presented such data. In the abstract, we wanted to convey the message that there is an increased frequency of LRV1 in hybrid parasites, and that such interactions are known (based on previous studies) to play an important role in disease severity. As the latter was already stated in the second sentence of the abstract ("In pathogenic protozoa, the presence of endosymbiotic viruses has been linked to an increased risk of treatment failure and severe clinical outcome.") and to avoid further confusion, we revised the final sentence to: "Our results suggest that parasite gene

flow and hybridization increased the frequency of viral endosymbiotic interactions, a process that may change the epidemiology of leishmaniasis in the region.”

We acknowledge the ongoing discussion on the relationship between LRV1 and disease severity, and our paper provides new and important insights that are relevant to investigate further within a clinical setting. As reported by our Peruvian colleagues, we are aware that there is an increase in drug treatment failures in the region of Huanuco where hybrid parasites are more frequently identified, although there is currently no epidemiological data to confirm this observation. Based on our data, we hypothesize that this increase may be due to an increase in LRV1 prevalence and diversity linked to parasite gene flow and hybridization. We are currently developing a sample enrichment technique that would allow us to sequence the parasite and viral genomes directly from clinical samples to avoid the inherent bottleneck when culturing parasites, which is important if we want to establish a direct link with clinical outcome and drug treatment failure¹⁰. Such a technique will allow us to investigate some of the questions raised by the reviewer in future research.

Reviewer #2:

The manuscript by Heeren et al is an interesting study of the genetic structure of *Leishmania braziliensis* in Peru and Bolivia coupled with the analysis of (*Leishmania* RNA virus 1) LRV1 distribution. The broad range of applied methods and the decent amount of analyzed isolates undoubtedly deserve credit. However, the manuscript has a number of important flaws and needs a thorough revision.

We thank the reviewer for giving credit to our extensive work and for the detailed revision of our paper. We believe that this has helped us to improve the quality of our manuscript.

1) There is a wrong positioning of the manuscript and disbalance of the content. In general, it appears that there is one study nested within another. In principle, there could be two different manuscripts. According to the title, the study should be focused on the increased prevalence of the virus in hybrids. However, most of the voluminous text is devoted to the genetic structure of *L. braziliensis*. Moreover, the main conclusion of the manuscript that has been put to the title is not even properly presented, The authors provide just the percentages of LRV1-bearing isolates for different genetic groups of the studied parasite and claim that there are significant differences without applying any statistical analysis for that. I am embarrassed by the necessity to provide an advice to researchers with an extensive expertise in statistics. However, this important message looks very weak. I believe that at least a simple Chi-squared test for proportions, as proposed by Campbell , 2007 is needed for the comparison of the two main groups: unmixed vs hybrid isolates. If the authors believe that their main finding concerns this

fact from the title, a reader would expect a more extensive discussion of this phenomenon, which is actually absent. The abovementioned disbalance can be diminished by decreasing the amount of text in the "populational" part by removing unnecessary parts related to methods and/or supplementary results. For example, the very first section "Natural genome diversity of Lb parasites" is mostly redundant. I would also suggest to think about reformulating the title to mention there that the study includes the analysis of the populational structure of *L. braziliensis*.

While our study contains sufficient material for two different manuscripts (one on *L. braziliensis* and another one on LRV1), we are convinced that the uniqueness and strength of this manuscript lies in the joint analyses of parasite and viral genomes to understand LRV1 diversity and dissemination in Leishmania populations. To this end, a detailed analysis of *L. braziliensis* population genomic structure is imperative before we can understand its impact on LRV1 evolution. However, we understand the concern of the reviewer regarding a disbalance of the content, and as such, we have restructured the results and discussion section following the reviewer's suggestions (see also below). We also agree that the title of the paper focuses only on one of the outcomes of our study, a concern that was also raised by reviewer 1. We therefore suggest a new title that better captures the breadth of our work: "Diversity and dissemination of endosymbiotic viruses in zoonotic Leishmania parasites". Finally, we acknowledge the need to better highlight and strengthen the link between parasite hybridization and viral prevalence, diversity and spread by including additional statistical analyses as suggested by reviewers 1 and 2. The new version of the manuscript now includes the following additional tests:

- 1. A Kruskal-Wallis test accompanied by pairwise Dunn's test to assess the differences in the geographic ranges between the parasite groups (lines 240-243; supp. table 8). The outcome of these tests show that hybrid groups were geographically more widespread compared to the ancestral populations.***
- 2. A Chi-squared test, as suggested by the reviewer, for comparing viral prevalences between the ancestral populations (PAU, HUP, INP) and the hybrid groups (ADM, UNC) along with Fisher's exact test to compare the viral prevalences between all the parasite groups in a pairwise manner (lines 371-374 Fig. 8; supp. table 13). Here we adopted Fisher's exact tests as some parasite groups had low sample sizes (<5). We also calculated the species (viral lineage) richness and Shannon diversity to test for differences in viral lineage diversity between the parasite groups (lines 381-382; supp. table 13). These additional tests and calculations clearly show that the hybrid groups, especially ADM, not only have a higher viral prevalence but also harbor a higher viral diversity compared to the three ancestral populations.***

- 3. A reconstruction of the spatial dynamics of LRV1 using BEAST v.1.10 to examine the dispersal history of LRV1 and assess the influence of each viral lineage on overall LRV1 diffusion rate (lines 334-339; Fig. 7B,C; supp. table 12). This revealed that the most prevalent viral lineage (L1) showed a much higher dispersal rate compared to the other viral lineages. It is also the lineage that is frequently associated with hybrid Lb parasites in tropical rainforests and surrounding ecological regions. This suggests that the spread of L1 might have been mediated by parasite gene flow and hybridization out of the tropical rainforest regions.***

Altogether, we provide additional statistical tests that show a link between parasite hybridization and the prevalence, diversity and spread of LRV1 in the region. As requested by the reviewer, we have significantly extended the discussion section to include a more lengthy interpretation of our results. We hope that the reviewer agrees with us that the additional analyses and the restructuring of the text has significantly improved the quality of our manuscript.

2) As mentioned above the whole text is too voluminous even though a proper discussion is absent. Moreover, many of the results mentioned in the text are illustrated by only supplementary figures. The authors should think, whether these results are really not important and then, maybe move them to supplementary Results (see above), or show some of the figures in the main text. Of note, the journal allows up to 10 figures, but only 5 are included into the main text. In addition to what is suggested above, the authors should also reorganize the text in such a way that description of the existing *L. braziliensis* groups is not interrupted. In fact, the groups are first mentioned in the sentence starting on the line 187. The authors allegedly identified all of them by the ADMIXTURE/fineSTRUCTURE analysis. However, on the Fig. 1B UNC is not labelled (and apparently does not form a solid group) and the isolate LC2043A1 is not classified (ADM group?). Apparently, the definitive separation of the six discussed group became available only in the results of the PCAdmix analysis mentioned on the line 238. Why they are separated by other analyses remains obscure. This creates a big mess in a reader's head.

We followed the reviewer's comment on the volume of the main text and attempted to shorten several parts. In doing so, we also made sure that the key messages come across stronger and less ambiguous. The main text now includes 8 main figures instead of 5. We believe this indeed improves the flow of our manuscript and thank the Reviewer for the valuable suggestion.

3) The section concerning "co-divergence" between *Leishmania* spp. and LRV1 was apparently added to the manuscript to somehow compensate the abovementioned disbalance in the

analyses of Leishmania and LRVs. However, it look redundant, since it is not logically connected with the other parts of the manuscript and, what is even more important, does not bring any new knowledge. The cited paper by Cantanhêde et al., 2018 has been already devoted to this issue and the new analysis despite the inclusion of additional sequences did not show anything new. The subdivision of LRV1 into lineages according to species has been demonstrated in the cited paper. Of note, this is only partially true, since the group of LRV1s from *L. guyanensis* show a very deep divergence and is non-monophyletic due to an isolate from *L. shawi*, which is nested within. In any case the analysis in Cantanhêde et al., 2018 is more comprehensive, since it considers not only to species of Leishmania, but also to geography.

We agree that the Cantanhede et al. (2018)⁴ study provides important prior work on this issue, but we believe that a major limitation is that the species-dependent structuring is based solely on partial LRV1 sequences. The paper of Cantanhede et al. (2018) does not include Leishmania genetic data, and as such, cannot formally test for co-phylogenetic signals⁴. In our work, we attempted to show this topological pattern of co-divergence by reconciling both virus and parasite phylogenies. We believe it is important to include these analyses because they reveal a clear co-phylogenetic signal at the interspecific level, while the remainder of our work shows a much weaker co-phylogenetic signal at the intraspecific level (i.e. within *L. braziliensis*-LRV1 from Peru and Bolivia) caused by prevalent horizontal transmission of LRV1 due to parasite hybridization. We have now repositioned and shortened this section in the manuscript so that it is more logically connected to the rest of the paper (lines 278-293).

4) The section named as discussion does not actually discuss the data, but repeats some of the statements from Results, which, in turn, do not represent pure results, but a mixture of results and discussion seasoned with a bit of methods. In a revised manuscript a proper discussion will be needed, which will consider also some results, which, for some reason were not of much use. I meant the inference of paleolandscapes, which should be considered for the explanation of ancient contacts between the three maternal populations of *L. braziliensis* described in the manuscript. It appears that the authors had something like this in mind, but then this was omitted. Of note, the claiming that "suitable habitats have been relatively stable over the past 130,000 years" does not correlate with the obtained results. The Fig. S10 shows that in the past the suitable areas were larger and that the present-day "islands" of optimal environmental conditions previously had contacts.

We have restructured the results section and lengthened the discussion section following the reviewers' suggestions. We also included a sentence in the discussion on the inference of paleolandscapes (lines 431-434). The main reason we included these analyses was to demonstrate that the suitable habitat for *Lb* has been subject to changes over the past

150,000 y, including major contractions during the Last Glacial Maximum. We believe that such forestation changes due to climatic cycling has promoted the diversification of Lb.

Now some medium-to minor concerns (the list is not comprehensive!):

1) The language of the manuscript needs improvement. There are some mistakes (e.g. "where" instead of "when"(line 80); missing articles; "on the European Nucleotide archive" and "on Genbank" (should be "in", lines 581 and 583); etc). In addition, the language is stylistically poor, with the same words used in one sentence up to three times. The abbreviation Lb for *Leishmania braziliensis* is a "mauvais ton" and must be avoided in the text. Although it may appear that the authors use it to shorten the sentences, in fact it is not true, since usually this abbreviation appears in a collocation "Lb parasites", which is approximately of the same length as "*L. braziliensis*", but reads very awkward. In most cases it is even not needed to mention the species, since the overwhelming part of the text is related only to this single species. Therefore, it is not needed to specify, whose genome, groups, populations, etc. are meant. There are cases of incorrect usage of terms, repetitive sentences, etc. See some additional examples are given below.

We are not entirely sure we follow the arguments of the reviewer. For instance, in the last sentence of the introduction we state that "Our previous work in Peru has shown that LRV1 was present in >25% of sampled Lb parasites, ..." Replacing the "sampled Lb parasites" by "sampled L. braziliensis" would render this sentence incomplete. On many other occasions, we use the abbreviation "Lb" without the word "parasite", which does reduce the length of the text. We also believe it is important to clearly and consistently state whether we are referring to the parasite or the virus, and whether we are referring to L. braziliensis or another Leishmania species. This is because there are sections dealing with both the virus and the parasite, and there are sections dealing with multiple Leishmania species. Hence, to be consistent and clear from the start, we introduced the abbreviation "Lb" to avoid any confusion.

2) The genetic groups discussed in the text must be referred as "groups", when listed, not as "populations" plus "groups". I understand, that the three of the groups represent maternal populations, by hybridization of which the other three appeared. This must be properly explained and the terms should be used carefully.

We agree that the different groups were inconsistently referred to throughout the text. We revised the manuscript and consistently used the same description for the groups. In the revised manuscript we refer to PAU, HUP and INP as the 'ancestral populations' and refer to ADM, UNC and STC as 'hybrid groups'.

3) Line 51: totiviruses > Totiviridae first mentioning should be with the Latin name).

Changed accordingly.

4) Line 53: yeasts are also fungi

The use of 'yeasts' is removed from the text.

5) Line 53: "closely related viruses identified in almost all genera of yeasts, fungi and protozoa" Seriously? In almost all genera of fungi and protozoa???

We understand the confusion in this paraphrasing of Bruen (2000) ¹¹. We meant that closely related viruses were identified in almost all genera of fungi and protozoa that were systematically studied at that time (2000). We revised the sentence to avoid further confusion.

6) Line 60: Family > family

We decapitalized the 'f' in family.

7) Line 64: " virus was most likely present in the common ancestor of Leishmania" – this is an obsolete view. The majority of Leishmania spp. do not have viruses. Some of those that have LRV2 (rare strains L. tropica and L. infantum) acquired it recently (the sequences are identical!) by lateral transfer from sympatric L. major. The situation with LRV1 also suggests at least such event between L. guyanensis and L. shawi. In addition, there are LRVs from monoxenous Blechomonas spp. Thus, there were several lateral transfers in the evolution of this viral genus, therefore there is no reason to assume its inheritance from a common ancestor.

It is true that the Leishmania-LRV co-evolution paradigm has been hypothesized three decades ago based on phylogenetic analyses that showed the co-divergence of LRV1 and LRV2 with L. (Viannia) and L. (Leishmania), respectively. We don't agree that this is an obsolete view, because there are many recent studies that confirmed the co-phylogenetic history between Leishmania and LRV ^{6,8} and that have shown species-dependent clustering in LRV1 ^{1-5,12}. A recent review including a comparative analysis of Trypanosomatid and LRV phylogenetic trees supported a long history of coevolution between Leishmania and LRV ¹³. As correctly indicated by the reviewer, there are also reports of recent horizontal transfer between different Leishmania species, but that does not refute the paradigm that LRV was present in the common ancestor of Leishmania prior to its divergence into the subgenera L. (Viannia), L. (Leishmania) and L. (Sauroleishmania). Similarly, the observation of LRV in Blechomonas spp. does not refute the co-evolution paradigm; in fact, the authors of that work concluded that a common ancestor of LRV most likely infected Leishmania first and was acquired by Blechomonas by horizontal transfer.

To provide more nuance to our sentence on line 64, we rephrased it to: "Despite reports of horizontal transmission, phylogenetic studies suggest that the virus was most likely present in the common ancestor of Leishmania prior to the divergence of these parasites into different species around the world".

8) Lines 75 and 82: these sentences are virtually the same.

We thank the reviewer for pointing this out. We have removed the sentence on line 82 and revised the sentence on line 75.

9) Line 95 (although this must not be in the main Results): specify that this concerns not all samples, but the majority of them.

We think there must be a misunderstanding about the line number, because we are not entirely sure what the reviewer is referring to.

10) Lines 115, 117: chromosomes cannot be diploid, this word refers to the state of the whole genome

We changed "diploid" to "disomic" in both instances.

11) Line 118: revise the sentence. The genomes were not entirely diploid if one of the chromosomes had a copy number other than double.

The sentence was revised and relocated to the heading of supplementary figure 2.

12) Line 123: " Variation in chromosome 123 and gene copy numbers was not associated..." – No data supporting this statement are provided. Either remove the sentence, or confirm it.

This sentence was omitted in the new version of this paragraph.

13) When you refer to figures, always mention a particular panel.

We revised the references to the multipanel figures in the main text accordingly.

14) Fig. 1A: in general, it is virtually useless, since does not show much. It can be either removed or transferred to Supplementary figures. In any case it is unclear why geography is shown only for some isolates and, moreover, why for most cases departments are indicated (Cusco, Cajamarca, etc), but for one there is the name of province (Huancapallac). The colouring of the branches is weird. Why black is used not only for ADM, but also for some internal branches.

Such a network is commonly used and shown in papers on population genomics of protozoan parasites. There is plenty of information that can be deduced from this network, such as for example the long branches that indicate prevalent recombination, the terminal clustering of

near-identical genomes that point to clonal reproduction and the grouping of the three ancestral populations. We therefore urge on keeping panel A. As stated in the Figure legend, we highlight the geography and numbers for the seven groups of near-identical genomes. But these are less relevant in the new version of the paper and are given anyway in a supplementary table, so we have chosen to omit these geography names from the network. The coloring of the terminal branches match the coloring of the parasite groups as shown in panels B and C, while the colors of the internal branches were left black. We added a sentence in the Figure legend to avoid further confusion.

15) Fig. 1B must be larger (full-size) to make the isolate names readable at the print size. I would also suggest to mark groups not by rectangles, but by colouring branches using the same palette as on the left. This is especially important given that it is difficult to delineate there the UNC group and label the LC2043A1 isolate.

We changed the font size of the isolate names so they are more readable and we have marked the groups by coloring branches, following the reviewer's suggestion.

16) Fig. 1C. is also rather small, consider enlarging it. In addition, the circle with 24 isolates appears to have a pink component. If this is not a mistake, then as judged by this panel this one isolate from Cusco, plus two from Junin, one from Ucayali and one from Loreto sum up to five total UNC isolates, not four as mentioned in the text. Correct this discordance.

This was an older version of the figure where LC2043 was wrongly coloured as UNC. We changed it to black now to show that it belongs to the ADM group.

17) For the convenience of readers, provide the classification of isolates into the six groups in the Table S1.

We agree that is useful information for the reader and included this in the mentioned supplementary table.

18) Lines 128-129 (legend): the information is false, since not all groups could be inferred using ADMIXTURE and fineSTRUCTURE. Only subsequent PCA-based analyses provided this classification.

We added "PCAdmix" to the sentence so it is clear the full classification has been based on all three analyses.

19) Line 132: "inferred at $K = 2$ or $K = 3$ populations": replace with "assuming 2 or 3 populations" for simplicity. Also simplify the text on the line 191. There is no need to use the K letter here, especially given that in ecology it may have different meanings.

It is common in population genomics studies to use the K within the context of ADMIXTURE analysis. We prefer to keep the K as it is a key parameter in model-based ancestry estimation and we believe the risk of confusion within this context is low.

20) Line 143: "It is clear" to whom? That is a bad style of writing.

We relocated this paragraph to the discussion section and changed this part of the sentence to: "It has been postulated that..."

21) Lines 144-147: The essence of the debate is not explained. The text reads like recombination is one of the modes of reproduction. Of note, this paragraph and the next one belong either to introduction or to discussion. Alternatively, the authors can decide, that they have a joint Results and Discussion section.

This paragraph was moved to the discussion section, where we explain the essence of the debate more clearly.

22) Lines 173-175: ungrammatical sentence.

This sentence was moved to the discussion section. The sentence was also revised slightly to make it grammatically correct.

23) Lines 178-179: ecotypes are entities occupying distinct ecological niches, so there is kind of tautology in this sentence.

We removed "that are each associated with particular ecological niches" from the sentence. This part was also moved to the discussion section.

24) Line 184: the abbreviation has been already explained

We removed "linkage disequilibrium" and retained only the abbreviation.

25) Line 185 and elsewhere: Present instead of Past tense (identify > identified).

Changed accordingly throughout the text.

26) Lines 191-192: "population" used thrice in a single sentence!

We revised the sentence to decrease the number of times "population(s)" was used.

27) Line 195: why only for ADM details are provided?

This was the most densely sampled group of isolates showing mixed ancestry patterns and also revealed most of the LRV1 associations.

28) Lines 196-215 (two paragraphs) must be moved elsewhere to separate this analysis from those focused on the genetic groups.

We have restructured the results section following the reviewer's suggestions and hope that the reviewer finds the paper more readable now.

29) Lines 203-206: repeated collocations in one sentence. Revise it and somehow explain to the reader, that the values are not additive, since 27,3% includes also the combined effect of the two factors.

We restructured the paragraph, removed the collocations mentioned by the reviewer, and included an explanation stating the strong confounding factors in the RDA models.

30) Line 209: revise the sentence. This analysis cannot say what habitats are suitable for a given species. It just predicts the spatial distribution of these habitats. So consider revising as "predicted putative suitable areas".

The sentence was revised as follows: "...predicted the putative suitable habitat distribution..."

31) Line 210: LIG – unexplained abbreviation

The abbreviation (Last Interglacial period - LIG) is now introduced in the sentence.

32) Line 210-213: sentence is ungrammatical and with repeated words

The sentence was revised.

33) Line 213: it is unclear whether savannah is considered as an unsuitable habitat (for some reason not even labelled on the map). However, STC isolates were obtained from tropical savannah.

We apologize for the lack of clarity in this part. We do consider tropical savannah as a more unsuitable habitat, which we now explicitly added to the text. We would like to note that "tropical savannah (Aw)" is labeled on the map in figure 2b. Please note that the focus of these environmental association analyses was on the three maternal populations and not on the admixed parasite groups.

34) In general, the text in this paragraph (Lines 208-215) must better explain what exactly the analysis is for.

We have rephrased some sentences and hope that our message is better conveyed now.

35) Line 217 (legend): explain that the maps show only maternal populations (carefully use the terms), but not hybrids.

We made slight changes to the figure caption of figure 4 (originally figure 2) following the reviewer's suggestion.

36) Fig. 2. In principle, it is unclear, why hybrids are not shown. Is not their distribution important? I do not think so. Moreover, I believe that the reconstructions of paleolandscapes from Figure S9 must be also shown here, at least with one of the rm values, although there should be no problem with space here. And provide their description in the text.

As the section's subtitle states, the analyses focus on showing that the ancestral populations (HUP, PAU and INP) are not only geographically separated, but also environmentally. We did this by highlighting that these populations are limited to pockets of tropical rainforest in the region and are fragmented by the tropical monsoon/savannah regions on one hand and the Andean ecoregions on the other hand. Within this context, we believe it is more intelligible to only show the distribution of the ancestral populations, and not the hybrid groups.

Concerning Figure S9, we prefer to show the results from the RDA analysis instead as it receives more attention in the main text.

37) Line 226: populations vs groups (see above).

The sentence on this line is revised.

38) Lines 235-237: this conclusion is not justified. Not all of these haplotypes could be inherited. Apparently III and IX were inherited, while X, XII, and XIV-XVII could evolve afterwards.

We believe the conclusion is justified because the presence of multiple haplotypes in the hybrid groups does suggest that these parasites are descendants from multiple hybridization events. Based on our data, there is no reason to assume the opposite, namely that the hybrids are descended from a single hybridization event. Also, the PCAdmix revealed complex patterns of mixed ancestry that can only be due to multiple cycles of hybridization and/or backcrossing. The presence of multiple mitochondrial haplotypes corroborates our findings at the nuclear level.

39) Lines 238-239: individuals > isolates

Changed accordingly.

40) Line 241: 92.9-99.4% of what?

We agree with the reviewer that it was not entirely clear what these percentages meant. We revised the sentence as follows: "While the control samples were assigned to their respective populations (92.9-99.4% of these haplotype blocks), the admixed individuals showed mixed ancestries between PAU (21.1-73.1%), INP (14.3-36.7%) and HUP (0.1-54.4%)."

41) Line 257: revise the sentence to make it clearer

This section was revised in the new version of the manuscript.

42) Lines 262-264: Why median distances? Given the uneven sampling density they are more than confusing. Apparently, the distances are needed to claim that " hybrid parasites are ... more widespread". Then it is better to provide the maximal distance, or a range of distances between sampling locations, where a particular group was detected.

We agree that the use of median distances here does not properly concur with our conclusion and therefore included the range of distances per parasite group as suggested by the reviewer. In addition we also tested the differences between these geographic ranges by means of a Kruskal-Wallis test and pairwise Dunn's test.

43) Lines 265-267: Why the authors do not consider contacts between the maternal populations in the past, when the suitable habitats could be connected to each other?

These lines were removed In the new version of the manuscript.

44) Fig. 3A: explain why the isolate LC2043 was classified as ADM. It looks isolated here and on the Fig. 1B. Also explain why the number of data points does not correspond to the number of isolates. For example. UNC should have four (or five? (see above)) isolates, but on this graph there are 7 points for this group.

Regarding the ancestry of LC2043: while fineSTRUCTURE and ADMIXTURE seem to group LC2043 within the PAU population (Figure 1), closer investigation of the haplotype co-ancestry matrix revealed that LC2043 shares ancestry profiles similar to those observed for the ADM parasites (Supp. Figure 7). In addition, PCAdmix confirms that the ancestry of LC2043 is uncertain as it cannot be clearly grouped with the PAU population. Our interpretation is that LC2043 has an ancestry similar to the ADM parasites, but that backcrossing with PAU parasites has diluted its hybrid profile.

We believe that the confusion regarding the number of isolates and the number of data points is because PCAdmix works with phased haplotype data. Therefore, each isolate is represented by two data points, one for each haplotype. Where the reviewer observed 7 points, there are in fact 8 points that are just really close to each other in the PCA space. We revised the figure caption to clarify that the PCAdmix scatter plot depicts points of parasite haplotypes.

45) Line 276: within what group of hybrid parasites? There are three of them

We do not see the need of specifying this in more detail in the subtitle. The subtitle is meant to provide the reader with the key message, which it portrays adequately. The associated paragraph brings the clarification and substantiation of the subtitle.

46) Line 281: dsRNA, not total

We see the point of the reviewer that the use of 'dsRNA' instead of 'total RNA' is more appropriate as not all RNA was present at the time of sequencing the RNA extracts. We revised this formulation according to the reviewer's suggestion.

47) Most of the text in the first paragraph on the page 10 is redundant. These are mostly technical details, that could be transferred to methods or supplementary results.

This paragraph is a neat summary of the breadth of work that has been done to recover high quality viral genomes from total transcriptome data. Most of the work has been described at length in the corresponding supplementary results section. We believe it is important to keep the information listed in this paragraph, as it summarizes i) the numbers of genomes that were successfully recovered from the LRV+ Lb parasites, ii) the observation that there are viral co-infections in two Lb parasites, and iii) the fact that we were able to recover high-quality viral genomes at sufficient read depth (31X-868X) despite that viral sequences only constituted a fraction of the total number of reads.

48) Line 285: "0.04% (0.004%-0.1%)" > "0.004-0.1%"

Changed accordingly.

49) Line 290: "covering the full-length coding sequence" – false, some of them (the shortest ones) contain only partial CDS

We changed "full-length" to "near-full-length".

50) Line 303: the procedure/software for the homoplasy index calculation is not mentioned in Methods

The software that was used to calculate the PHI is mentioned in the method section of the manuscript on line 649. In addition, we added the reference of the PHI calculation directly in the sentence on line 311.

51) Line 304: "t=5.72" - What is this value?

We acknowledge the provided information was too limited to fully understand what was meant in this section. We performed a correlation test which highlighted the positive association of the number of detected LRV lineages with parasite sample size. We provided more appropriate information on the correlation test to make this more clear in the text and in supplementary figure 14.

52) Line 304: twice "sampled" in the sentence

We removed one occurrence of “sampled” in the sentence.

53) Lines 305-311: discussion of the number of lineages for such minute geographical groups, many of which contain a single LRV+ isolate is not much useful.

As we are the first to examine LRV1 population structure within Peru and Bolivia in such detail, we believe it is important to provide insight into the geographical distribution of LRV1 in this region in just a few lines.

54) Line 318: " The distal position of the two Bolivian L7 strains within a clade of Peruvian viral lineages" replace simply with "This"

We believe there was a slight misunderstanding and we have rephrased this sentence to point out that the Bolivian L7 strains are distal within a larger clade of Peruvian viral lineages (L2, L3, L4, L5 and L6). We hope that this better conveys our message.

55) Line 320: " all early-diverging lineages (L2-L9)" > "all lineages but L1". All lineages cannot be early-diverging.

We revised this part of the sentence as: “... nearly all lineages (L2-9)...”.

56) Line 320: "was introduced" Apparently "must be "recently introduced"

The sentence was revised accordingly.

57) Line 323: "temperate climate" – This is not labelled on the map. IN addition, it is not possible to see on this figure, where the isolates (or locations) belong, because the areas are tiny as compared to the size of the circles.

We added the label for the Cwb climate region in Figure 6 (previous Figure 4). The reviewer is right that it is hard to infer the climate region for each isolate based on this map, but the main goal of the map in Figure 6 (previous Figure 4) is to provide a general overview of the distribution of LRV1 lineages in Peru and Bolivia. We now also included an additional plot in Figure 7A that shows more clearly the distribution of all LRV1 lineages per climatic region.

58) Line 331: "50 Lb isolates". Must be 51 (10+20+21)

We apologize for the minor mistake. After a thorough revision of the manuscript and linked to the comment of the reviewer concerning isolate LC2043, the mistake was situated in the total number of isolates in PAU which should be 19 instead of 20. So, the total sum was correct (19 + 21 + 10 = 50).

59) " These observations strongly suggest" – No, this is not true. They cannot suggest this (moreover, strongly), since the CUM65 isolate (definitely from a single population) contains two

different lineages! I would understand if you claimed that in each of these populations one viral lineage dominates (do not forget about an insufficient sampling).

This section was revised in the new version of the manuscript.

60) Line 337: "contained almost all viral lineages (L1, L3-L9)" > "all viral lineages except for L2"

The sentence was revised according to the reviewer's suggestion along with an additional correction. It is revised as: "... all lineages except for L2 and L8 ...".

61) Line 339: gene flow is irrelevant here

We changed "gene flow" with "dispersal" as it fits better with the message.

62) Figure 4: numbers at branches are not explained

We apologize for the negligence. The numbers represent the branch support values based on 100 bootstrap replicates. This was stated in the method section but, we have now also added this to the figure caption of figure 4.

63) Line 358: repeating the unjustified statement from the introduction (see above)

This sentence was moved to discussion. Please see also our answer to the reviewer's question 7.

64) Line 360: why "relatively" if there are identical sequences? Specify, that this concerns LRV2 (No more comments on this section, which should be removed)

This section was indeed moved to Discussion and changed following the reviewer's comments.

65) Line 400: "two species" are not specified

We revised the sentence so that the two species are specified: *L. braziliensis* and LRV1.

66) Line 402: highly heterogeneous as compared to what? How do you know, where LRV1 predominantly evolved? It is better to mention, where you detected it, but this will not be the exhaustive list of locations, especially given that this virus occurs in other countries and other species of Viannia.

As stated in the results section, all viral lineages but L1 were found in the tropical rainforest. Also, the phylogenetic tree in Figure 4 is rooted and shows that L1 is one of the more late diverging clades. These observations indicate that LRV1 evolved predominantly in the tropical rainforest. Nevertheless, this statement was removed in the new version of the manuscript. *Lb* and LRV1 show extensive genetic diversity and population substructure, and are thus genetically heterogeneous. We believe this statement is clear.

67) Lines 420-422: it is not possible to judge from the obtained data whether the hybridization was recent or ancient. The only conclusion that can be made at the current moment is that none of the hybrids was F1.

The discussion of the new version of our manuscript includes a more extended argument as to why we believe that these hybridization events occurred over the past century, and are thus relatively recent.

68) Line 441: 29 is not a half of 55. Specify the exact number.

We changed "half" to "more than half".

69) Line 446: concentration of FBS? Any antibiotics supplemented?

The FBS concentration in the liquid medium was 20%, which we added to the main text. No antibiotics were supplemented to the cultures.

70) Lines 453-455: awkward sentence with repeats

We revised the sentence slightly.

71) Line 460: LRV1 was not sampled, it was detected in some isolates

The sentence was revised, so that it states that LRV1 was detected in isolates from Madre de Dios, Junin, Cajamarca, Huanuco, Loreto and Ucayali.

72) Line 462: do not repeat the cultivation conditions

We brought the LRV1-positive isolates twice into culture for different types of nucleic acid extractions. Hence, we reported our methodology correctly.

73) Line 449 and 464: the number of cells collected?

This information was added.

74) Line 465 (reference 58): why this reference, which does not provide any additional details, but refers to an earlier paper?

The earlier paper (Grybchuk et al. 2018⁸) also refers to an even earlier paper (Chomczynski et al. 1987¹⁴). We chose to refer to Kleschenko et al. (2019)¹⁵ as this paper simply represents the methodology of the dsRNA extraction which we also summarized in our method section. We believe it is appropriate to acknowledge the work that served as an inspiration for the methods we adopted.

75) Line 467: the name of the Zymoclean kit?

Zymoclean Gel RNA Recovery kit. The kit's name was added to the text.

76) Line 470: number of reads generated?

We included the average number of reads that was generated (35,665,319 paired end reads).

77) Line 473-475: revise the sentence

The repetition in the sentence was removed.

78) Line 475: "the reference genome" > "the reference genome assembly"

The sentence was changed according to the reviewer's suggestion.

79) Line 477: specify the ID of the assembly (there can be more than one)

Added as requested.

80) Line 486: "The final set... were" – ungrammatical

We changed "final set" to "final sets".

81) Line 491: reference for SAMtools

Reference added.

82) Line 499: " A NeighborNet tree" > "A Neighbor-Net network" or " A Neighbor-Net graph"

Changed accordingly to "Neighbor-Net network".

83) Line 501: why the results are shown only for K=2 and K=3?

K=3 and K=2 had the lowest cross-validation errors with the same order of magnitude as K=1 (K=1 has the actual lowest CV error). This prompted us to show the structuring of Lb on these two levels. Combining these results together with fineSTRUCTURE and PCAdmix result in a well-founded description of the genomic structuring in Lb. We added this explanation of why we only focused on K=2 and K=3 in the methods section on line 591 and added a supplementary table with the CV error values for all ADMIXTURE models (K = 1-10).

84) Line 502: double "with". The sentence seems to be unfinished.

We thank the reviewer for noticing this and apologize for our negligence. The sentence has now been completed.

85) Line 504: "genetic ancestry" > "ancestry" (what else it can be, spiritual?)

Changed accordingly.

86) Line 505: delete "computationally"

Changed accordingly.

87) Line 507: "for finding ... for building" – revise

The part of the sentence was revised as follows: "... for finding the best tree topology."

88) Line 510: examined > assessed; delete "the"

Changed accordingly.

89) Line 514: "isolates ... isolated" – tautology; "year of isolation" > "year"

We changed "isolated" to "sampled", but kept the use of "year of isolation" as it is more specific.

90) Line 517: included into what?

We apologize for the lack of clarity. We replaced "included" with "calculated".

91) Line 542: specify the algorithm

We specified that we used the L-INS-i algorithm in MAFFT v.7.49.

92) Line 544: more inappropriate abbreviations of Leishmania spp. names...

Please see our response to comment 1.

93) "to 5189 bp ... and ... to 755 bp" – revise the sentence for clarity. Are these minimal lengths of sequences?

We added "trimmed to the minimal sequence length" to avoid confusion.

94) Line 550: Lines 547-551: revise for conciseness

We revised these sentences as requested.

95) Line 552: distances

Changed accordingly.

96) Line 563: added to what?

We changed this to "complemented our dataset with"

97) Line 564: the accession number is wrong, also specify the database (GenBank)

We thank the Reviewer for pointing this out. We discovered a typo in the SRA accession number and corrected the mistake. The specific databases were added to the text according to the reviewer's suggestion.

98) Line 566 – used software is not specified
The used R package was added to the text.

99) Lines 576-578: the sentence is vague
We changed this sentence and hope it is clearer now.

100) Data availability: submit the assembled maxicircle sequences and provide the accession numbers
We did not assemble maxicircle sequences but generated SNP genotypes after aligning the reads to the reference genome, which includes the 35 chromosomes and a complete mitochondrial maxicircle. This information was given in the Methods section.

101) Figure S11: why UNC is absent from the network?
This is an oversight. UNC was added now.

102) Figure S14, legend, line 45: numbers are on the left, not on the right and they show not the count of positive *L. braziliensis* isolates, but the numbers of LRV1 lineages (as judged by the text of the manuscript). This was not easy to find out...
We thank the reviewer for pointing out this mistake in the figure caption. We changed the text of the caption so that it matches with the text and figure.

103) Figure S15. No legend for colours. Why French Guiana is coloured? What are the supports (do not appear as bootstrap percentages)?
The coloring of the isolates is consistent with figure 4a and figure 5a. The supports are indeed in bootstrap percentages. This was added to the legend.

Reviewer #3:

This study by Heeren and collaborators analyzed the hybridization of *Leishmania v. braziliensis* (Lb) protozoan parasite favoring the propagation of the endosymbiotic viruses (LRV1). This an important epidemiological study for investigators toward the comprehension on how LRV1 could play an important role in the pathological development of MCL and the appearance of atypic phenotype of the disease. Use of ecological biotope modelisation could contribute to the monitoring of geographic distribution of *Leishmania v.* and LRV1.

The manuscript is very well written, concise, and clear, and the results are presented in a structured and organized manner. The introduction provides a clear and informative context regarding *Leishmania* and the endosymbiont virus LRV1.

We appreciate the positive feedback of the reviewer and thank the reviewer for the in-depth evaluation of the Ecological niche models that were generated in this study. Based on the comments and suggestions of the reviewer, we re-evaluated our part on the ENMs where we identified an inconsistency in the set up of the spatial projection of the past climatic data. Therefore, we decided to redo the analysis and reported a more thorough methodology and results of this analysis. We believe that this has helped us to improve the quality of our manuscript.

Comment and Suggestions

About the model,

An essential step for ecological niches modelisation consists of a statistical analysis toward the identification of all covariations of bioclimatic variables to select an optimal group of 19 variables necessary for the modelisation.

In their study, the authors did not mention this crucial step. This needs clarification.

We apologize for the improper description of the ENM variable selection. We revised the methodological part of our ENM analysis to better explain how variables were selected and how the ENMs and spatial projections were generated. In this regard we included an additional supplementary table (Supp. Table 18) depicting statistics on the abiotic variable selection and the model performances.

Add the graphs regarding the most elevated gain for environmental variables in the model for each Lb population and explain in results how they influence the model.

As requested, we added graphs depicting the models' ROC and elevated/lost gain for the environmental variables included in the models. These graphs are brought together in Suppl. Fig 19.

Why did the authors not use the data about missing species?

Modelisation of niches should be based on a combination of presence or absence of species permitting a better determination of environmental preferences.

We generated ENMs based on presence-only data against a background of 10,000 pseudo-absence points that were randomly generated. We agree that the use of true absence data results in more performant models. However, such data requires specific sampling designs meant for these types of analyses, which are often lacking in pre-existing sample collections. For this reason the use of pseudo-absence points can be adopted to mimic the absence of the species of interest in the study region.

Explain the number of presence monitoring that they used for the modelisation.

As per, "van Proosdij AS, Sosef MS, Wieringa JJ, Raes N. Minimum required number of specimen records to develop accurate species distribution models. *Ecography*. 2016 Jun;39(6):542-52 ».

The total number of unique presence points is 14 (Several isolates originate from the same location). We acknowledge that this is a low sample size and are aware that this might affect the accuracy of our environmental niche models. However, some approaches, e.g. Maxent, are much less sensitive to low sample sizes¹⁶. In order to increase the number of presence points and simultaneously account for the uncertainty around the sampling locations, we added some noise to all duplicated coordinate pairs by means of a jitter (factor = 0.01).

What they expect for appropriate habitat or potential distributions of Lb population regarding future bioclimatic variables?

We concur with the idea of predicting the potential future suitable habitat for Leishmania, although here the main goal of our analyses is to provide a possible explanation for the historical divergence of Lb in the region. Predicting Lb habitat suitability for different future climatic scenarios is part of an ongoing investigation where we are modeling the distribution of Lb and other L. (Viannia) species across South-America.

Does different Lutzomia sps could explain some of the distribution obtained?

The distribution of different vector species and host-vector communities are very likely to play a role in the observed distributions. We raised this possibility in the original discussion (line 407) and kept it in the revision (line 435).

Why not all viannia are found with LRV1 endovirus? To which extend, in comparison to Lviannia without LRV1, the MCL development is different? Lviannia without LRV1 will also in a very similar proportion lead to MCL upon initial cutaneous lesion... This could be further discussed.

There is still a gap in our current knowledge on the LRV-Leishmania co-evolution that should be further examined on the species level rather than the population level. At this point, our data indicates that genetic drift may lead to the loss of viral diversity in isolated parasite populations, which could explain why LRV1 is absent in some clonal and bottlenecked Leishmania species (this interpretation was added to the discussion section on lines 473-475). Regarding the role of LRV1 in human tegumentary leishmaniasis: LRV1 was shown to modulate the human host immune response including the NLRP3 inflammasome, which could contribute to an exacerbated skin pathology caused by LRV1-bearing Leishmania¹⁷. However, there are probably various factors (including human host, Leishmania parasite, LRV1 and environment) that contribute to the different clinical outcomes in tegumentary leishmaniasis.

References

1. Klocek, D. *et al.* Evolution of RNA viruses in trypanosomatids: new insights from the analysis of Sauroleishmania. *Parasitol. Res.* (2023) doi:10.1007/s00436-023-07928-x.
2. Nalçacı, M. *et al.* Detection of Leishmania RNA virus 2 in Leishmania species from Turkey. *Trans. R. Soc. Trop. Med. Hyg.* **113**, 410–417 (2019).
3. Saberi, R. *et al.* Presence and diversity of Leishmania RNA virus in an old zoonotic cutaneous leishmaniasis focus, northeastern Iran: haplotype and phylogenetic based approach. *Int. J. Infect. Dis.* **101**, 6–13 (2020).
4. Cantanhêde, L. M. *et al.* New insights into the genetic diversity of Leishmania RNA Virus 1 and its species-specific relationship with Leishmania parasites. *PLoS One* **13**, e0198727 (2018).
5. Zangger, H. *et al.* Leishmania aethiopica field isolates bearing an endosymbiotic dsRNA virus induce pro-inflammatory cytokine response. *PLoS Negl. Trop. Dis.* **8**, e2836 (2014).
6. Kostygov, A. Y. *et al.* Analyses of Leishmania-LRV Co-Phylogenetic Patterns and Evolutionary Variability of Viral Proteins. *Viruses* **13**, (2021).

7. Widmer, G. & Dooley, S. Phylogenetic analysis of Leishmania RNA virus and Leishmania suggests ancient virus-parasite association. *Nucleic Acids Res.* **23**, 2300–2304 (1995).
8. Grybchuk, D. *et al.* RNA Viruses in Blechomonas (Trypanosomatidae) and Evolution of Leishmaniavirus. *MBio* **9**, (2018).
9. Tirera, S. *et al.* Unraveling the genetic diversity and phylogeny of Leishmania RNA virus 1 strains of infected Leishmania isolates circulating in French Guiana. *PLoS Negl. Trop. Dis.* **11**, e0005764 (2017).
10. Domagalska, M. A., Barrett, M. P. & Dujardin, J.-C. Drug resistance in Leishmania: does it really matter? *Trends Parasitol.* **39**, 251–259 (2023).
11. Bruenn, J. A. Viruses of Fungi and Protozoans: Is Everyone Sick? *Viral Ecology* 297–317 (2000).
12. Hajjarian, H. *et al.* Detection and molecular identification of leishmania RNA virus (LRV) in Iranian Leishmania species. *Arch. Virol.* **161**, 3385–3390 (2016).
13. Cantanhêde, L. M. *et al.* The Maze Pathway of Coevolution: A Critical Review over the Leishmania and Its Endosymbiotic History. *Genes* **12**, (2021).
14. Chomczynski, P. & Sacchi, N. Single-step method of RNA isolation by acid guanidinium thiocyanate-phenol-chloroform extraction. *Anal. Biochem.* **162**, 156–159 (1987).
15. Kleschenko, Y. *et al.* Molecular Characterization of Leishmania RNA virus 2 in Leishmania major from Uzbekistan. *Genes* **10**, (2019).
16. Guisan, A., Thuiller, W. & Zimmermann, N. E. *Habitat Suitability and Distribution Models: with Applications in R.* (Cambridge University Press, 2017).
17. de Carvalho, R. V. H. *et al.* Leishmania RNA virus exacerbates Leishmaniasis by subverting

innate immunity via TLR3-mediated NLRP3 inflammasome inhibition. *Nat. Commun.* **10**, 5273 (2019).

REVIEWERS' COMMENTS

Reviewer #2 (Remarks to the Author):

The authors significantly improved the manuscript by restructuring it and adding further analyses. Now it looks more solid. However, the text still needs improvement and many figures do not have a sufficient resolution to make the text in them readable.

I have two main concerns:

1) The issue of taxonomic names' abbreviation. There are certain established rules of how Latin names of taxa should be written and abbreviated. I am not sure if the Journal is tolerant to ignoring that. I consider using Lb and Lg for *Leishmania braziliensis* and *L. guyanensis* absolutely unacceptable. I have previously tried to explain the authors that they do not gain much from such an approach, and they decided that this is a matter of discussion. However, it is definitely not! Therefore, replacements must be made throughout the text. However, the majority of instances does not need mentioning species, as I have previously tried to explain. The abbreviated form became for the authors so inconspicuous that they used it up to three times in a sentence.

2) The acquisition of LRVs in the common ancestor of *Leishmania* (L63-66). This concept was proposed in 1995, when the phylogeny and diversity of *Leishmania*, as well as the distribution of viruses were not properly known. At that time the genus was usually considered to comprise only the subgenera *Leishmania* and *Viannia* (*Sauroleishmania* was often considered a separate genus). More importantly, the early-diverging subgenus *Mundinia* with global distribution was not yet described. This simplistic picture inevitably led to the conclusion that since both *Leishmania* and *Viannia* have LRVs, the latter were acquired by a common ancestor (of all *Leishmania*). This was a conclusion based on the principle of maximum parsimony. However, no LRVs have been so far detected in *Mundinia*, suggesting that LRVs could be acquired after its separation from the rest of *Leishmania* (i.e. not in the ancestor of the whole genus). In addition, in other *Leishmania* spp., we observe a considerable potential to horizontal transfer of LRVs (not only to other species and subgenera, but even to *Blechnomonas*). This, in turn, suggests that LRVs also could be (I do not insist on that) acquired AFTER the split of Old World and New World lineages of *Leishmania*, but BEFORE the physical separation of the continents, followed by a horizontal transfer between these recently separated lineages. Even at the interspecific level such transfers are not so infrequent! None of the above arguments can fully exclude the possibility of what the authors stated. But it is not true that "phylogenetic studies suggest that the virus was most likely present in the common ancestor of *Leishmania*". This interpretation is not "most likely" and it cannot be directly derived from the latest phylogenetic studies. Horizontal transfers and absence of viruses from many species, including the divergent ones, demonstrates that evolution of LRVs cannot be interpreted using the parsimony principle. Thereby they significantly lower the likelihood of the hypothesis formulated by Widmer and Dooley. It still can be presented as a hypothesis, but with a proper explanation.

Minor comments:

L35-36: "Viruses... play significant role in the evolution of many organisms and ecosystems." Why not all, but many?

L40 and elsewhere: The collocation "*Leishmania braziliensis* parasites" (and the same in the abbreviated form) scattered through the text of the manuscript reads weird and ambiguous, since someone would first think about the parasites OF *Leishmania braziliensis*. In all cases it must be omitted. The authors argued, that some sentences may read unfinished without it, but this is not true. In such cases other (meaningful) words should be used, e.g. "isolates".

L40 and elsewhere: "endosymbiotic (dsRNA) virus". This is virtually tautology, since viruses are always endosymbiotic. Remove this collocation from the whole text.

L41: "ancestral parasite populations" – for the abstract this appears as unexplained concept.

L42: were > are (the tense differs from the previous sentence)

L44: Is it exactly the gene flow, or rather spreading of particular strains/lineages?
Spread > dispersal (?)

L45: "viral endosymbiotic interactions" – What is that? Remove "endosymbiotic" and rewrite to make it understandable. I assume that this is still about the viral prevalence, diversity and dispersal of the viruses. Then this sentence can be simpler. Just delete "increased the frequency of viral endosymbiotic interactions, a process that may", then the sentence will read more logically: "Our results suggest that parasite spreading and hybridization impact the epidemiology of leishmaniasis in the region."

L48: delete "virtually", unless you know exceptions

L48-51: The whole sentence is difficult to read because of the simultaneous use of the words "that", "which" and "such".

L49: Reads as "some of which" refer to eukaryotes.

L50: Why these biological roles are more important than others that are not mentioned here? Apparently the authors wanted to outline the effects that are of practical importance. Then this sentence should be revised accordingly.

L51: "Among the RNA viruses, the double-stranded RNA (dsRNA) Totiviridae" Reads awkward. I suggest: "The double-stranded RNA viruses of the family Totiviridae"

L55: delete "symbiotic" (see reasoning above)

L56: it is > it has been

L65-66:

L67: + Sauroleishmania

L78: it also circulates in dogs, which are not wild. I suggest to delete this word

L81: Here, for example, mentioning species is already redundant.

L82: another example: sampled Lb parasites > isolates
I will no more mention every instance of the studied species name and how to treat it.

L86-90: Simplify the text: "We sequenced genomes of 79 isolates of *Leishmania braziliensis* with known LRV infection status, which had been sampled during various studies on the genetics and epidemiology of leishmaniasis 22,29–33. The read coverage ranged from 35x to 121x (median value - 58x)."
It is not needed to mention here the reference genome.

L90-92: this sentence is fully redundant. It does not say anything about the data. Mentioning software here is unjustified.

L92-94: It is unclear why these three isolates were removed. Please, bring back from the previous version the text explaining that.

L95: unexplained abbreviation

L99-100: What are these other eukaryotic genomes? The provided reference concerns *Leishmania*, I suggest mentioning just this genus. Moreover, I doubt that parasites these exact values are expected. Make a separate sentence like: "This is in line with what is known for the genomes of *Leishmania* spp."

L105-106: It is unclear why only one curve from this graph is considered, while the three others show higher values. This must be properly explained.

L107: I am not sure, whether 0.2 is "around zero" in the scale from 0 to 1.

L112: differed at an average 9,866 fixed SNP differences > had in average 9,866 fixed SNP differences

L114: cluster > clustered

L118: and one lineage > and one of them

L123-124: "We identified three distinct ancestry components..." Specify the analysis that inferred this.

L140: at > with

L143: "three main parasite groups": You seemed to decide naming them as (ancestral) populations.

L146: Gray-scale > Gray scale

L152: the earlier described > the identified (otherwise it reads as if these populations were described in a previous work. Moreover, the word described is not suitable here, since they are not much described so far.)

L162-163: The differences consist in single nucleotide positions. Therefore these also could be random mutations, which occurred after the hybridization.

L166: to reference > to the reference

L172: hybridization, not admixture, if we discuss event(s). Admixture is the result of such event(s).

L179: Lb parasites > lineages

L211-212: to either of the explanatory components > to individual components

Figure 3: It would be more convenient if the values in the graph and in the text were in the same format, i.e. everything only as fractions (0.273) or percentage (27,3%).

L241-244: The values in parentheses are distances between the sampling points, where these populations were documented. Geographic ranges are not the same as ranges of distances.

L255: analyzed > included; as revealed by previous > as revealed in a previous

L268: Sequences > The assembled LRV1 sequences

L280: LRV1 genomes of > LRV1 genomes from

L281-283: Please, describe the topology of the tree. The position of LRV1 from *L. shawi*, breaking the clade of LRV1 from *L. guyanensis* must be mentioned. This apparently represents a rare random horizontal transition event, which nevertheless does not exclude the essential proportion of co-phylogeny between the other two *Leishmania* spp. and their viruses.

L284-285: mention the two species just once (not thrice (!) in the sentence)

L286-287: Dichotomy cannot be between something. Rephrase, or better unite this sentence with the next one, since actually both concern the same split.

L298: Co-divergence between > Co-divergence of

L300-301: " LRV1 clades of Lb are coloured according to the different viral lineages identified in this study (L1-L9)" > LRV1 sequences from *L. braziliensis* are coloured according to the viral lineages identified in this study (L1-L9).

Figure 5: please, mark the lineages L1-L9 on the tree to make it easier readable,

L303: monophyletic LRV1 clades > LRV1 clades (clades are monophyletic by definition)

L326: Viral sequences of the Bolivian Lb parasites > Viral sequences from the Bolivian isolates

L329: the L7 viral lineage > the former

L339-340: This is a repeat of the above information, delete the sentence.

L353: " The distribution is shown for LRV+ Lb isolates belonging to the nine viral lineages" *Lesihmania braziliensis* isolates cannot belong to viral lineages. Suggested revision: "The distribution is shown for the nine viral lineages"

L362: phylogeny > dataset

L374: Specify, that this is because of the low sample number for this group.

L388: less > lower

L415: dissemination > distribution; delete "endosymbiotic"

L416: Why these "here"? You cannot generalize these observations. Write directly about the objects of study.

Then tons of unnecessary "Lb" in the discussion...

L448: "cycles of hybridization events". Why cycles? It is not a cyclical process. Just hybridization events.

L455: was > has been

L456: "in strains of *L. (Viannia) species*" > "for several species of the subgenus *L. (Viannia)*" OR "for several *L. (Viannia) spp.*"

L456: and also > as well as

L458: clustered according to > clustered mostly according to (because this is not absolute)

L481: to the other viral lineages > to the others

L482: that parasite gene flow > parasite expansion (?)

L489-490: human migration, including movement of hemerophile reservoirs/hosts > migration of humans and hemerophile reservoir hosts

L491, 505: endosymbiotic (see above)

L493: " integrated into the Peruvian nation" Maybe "had poor contacts with the rest of the country"? Otherwise it reads absurd.

L497: Amazon regions > the latter

L501: replace comma with period and start a new sentence (from "This" instead of "which")

L533: on a HOMEM medium > in the HOMEM medium; also specify the manufacturer

L545: delete redundant "isolates"

L550: > Were grown as above for 2-3 weeks. (avoid repeating the same information)

L554: (NEB.) > (NEB)

L558: " NovaSeq platform" – specify the model of the instrument;
generating an average of > generating in average

L587: p-distances between what?

L596: with 50% as burn-in > with 50% burn-in

L642 and elsewhere: IQtree > IQ-TREE

L643: function > module

L710: on > in

L896-872: this is now a published paper

Reviewer #3 (Remarks to the Author):

Based on answers from authors to our queries, I found the manuscript acceptable as it is. They provide important information further supporting their study.
Not more to add.

Rebuttal - LRV paper

Revision #2

Reviewer #2

The authors significantly improved the manuscript by restructuring it and adding further analyses. Now it looks more solid. However, the text still needs improvement and many figures do not have a sufficient resolution to make the text in them readable.

We thank the reviewer for acknowledging the improvement of the manuscript after a thorough revision, as well as for their last comments on the manuscript.

I have two main concerns:

1) The issue of taxonomic names' abbreviation. There are certain established rules of how Latin names of taxa should be written and abbreviated. I am not sure if the Journal is tolerant to ignoring that. I consider using Lb and Lg for *Leishmania braziliensis* and *L. guyanensis* absolutely unacceptable. I have previously tried to explain the authors that they do not gain much from such an approach, and they decided that this is a matter of discussion. However, it is definitely not!

Therefore, replacements must be made throughout the text. However, the majority of instances does not need mentioning species, as I have previously tried to explain. The abbreviated form became for the authors so inconspicuous that they used it up to three times in a sentence.

Although this type of abbreviation is not uncommon in papers on protozoan parasites, we decided to change all instances of 'Lb' and 'Lg' and adopt the conventional way of abbreviating binomial species names, as requested by the reviewer.

2) The acquisition of LRVs in the common ancestor of *Leishmania* (L63-66).

This concept was proposed in 1995, when the phylogeny and diversity of *Leishmania*, as well as the distribution of viruses were not properly known. At that time the genus was usually considered to comprise only the subgenera *Leishmania* and *Viannia* (*Sauroleishmania* was often considered a separate genus). More importantly, the early-diverging subgenus *Mundinia* with global distribution was not yet described. This simplistic picture inevitably led to the conclusion that since both *Leishmania* and *Viannia* have LRVs, the latter were acquired by a common ancestor (of all *Leishmania*). This was a conclusion based on the principle of maximum parsimony. However, no LRVs have been so far detected in *Mundinia*, suggesting that LRVs could be acquired after its separation from the rest of *Leishmania* (i.e. not in the ancestor of the whole genus). In addition, in other *Leishmania* spp., we observe a considerable potential to

horizontal transfer of LRVs (not only to other species and subgenera, but even to Blechomonas). This, in turn, suggests that LRVs also could be (I do not insist on that) acquired AFTER the split of Old World and New World lineages of Leishmania, but BEFORE the physical separation of the continents, followed by a horizontal transfer between these recently separated lineages. Even at the interspecific level such transfers are not so infrequent! None of the above arguments can fully exclude the possibility of what the authors stated. But it is not true that "phylogenetic studies suggest that the virus was most likely present in the common ancestor of Leishmania". This interpretation is not "most likely" and it cannot be directly derived from the latest phylogenetic studies. Horizontal transfers and absence of viruses from many species, including the divergent ones, demonstrates that evolution of LRVs cannot be interpreted using the parsimony principle. Thereby they significantly lower the likelihood of the hypothesis formulated by Widmer and Dooley. It still can be presented as a hypothesis, but with a proper explanation.

We follow the reasoning of the reviewer and we are also pleased to read that the reviewer clearly states that "None of the above arguments can fully exclude the possibility of what the authors stated". To accommodate the concerns of the reviewer, we have rephrased these two sentences by removing the statement that "phylogenetic studies suggest that the virus was most likely present in the common ancestor of Leishmania". We now refrain from any statements regarding the timing of the acquisition of the virus. We also keep reference to the papers describing horizontal transmission as well as the papers suggesting co-evolution between Leishmania and LRV, but we do not want to discuss this in further detail in the introduction as it is not the purpose of our study.

Minor comments:

L35-36: "Viruses... play significant role in the evolution of many organisms and ecosystems."

Why not all, but many?

We are reluctant to use the word 'all' as we do not know whether viruses play an important role in the evolution of 'all' organisms.

L40 and elsewhere: The colocation "Leishmania braziliensis parasites" (and the same in the abbreviated form) scattered through the text of the manuscript reads weird and ambiguous, since someone would first think about the parasites OF Leishmania braziliensis. In all cases it must be omitted. The authors argued, that some sentences may read unfinished without it, but this is not true. In such cases other (meaningful) words should be used, e.g. "isolates".

The colocations were removed from the text as suggested by the reviewer.

L40 and elsewhere: "endosymbiotic (dsRNA) virus". This is virtually tautology, since viruses are always endosymbiotic. Remove this colocation from the whole text.

As requested, we avoided the use of "endosymbiotic viruses" throughout the text.

L41: "ancestral parasite populations" – for the abstract this appears as unexplained concept.
We omitted "ancestral" from the sentence.

L42: were > are (the tense differs from the previous sentence)
Changed accordingly.

L44: Is it exactly the gene flow, or rather spreading of particular strains/lineages?
Spread > dispersal (?)
The choice of word is correct here as we do mean parasite gene flow. This is not to be confused with the spreading of particular viral lineages.

L45: "viral endosymbiotic interactions" – What is that? Remove "endosymbiotic" and rewrite to make it understandable. It assume that this is still about the viral prevalence, diversity and dispersal of the viruses. Then this sentence can be simpler. Just delete "increased the frequency of viral endosymbiotic interactions, a process that may", then the sentence will read more logically:
As requested, we removed "endosymbiotic", and slightly changed the sentence. We insist to keep the rest of the sentence as the suggestion of the reviewer ignores the viral part.

L48: delete "virtually", unless you know exceptions
Changed accordingly.

L48-51: The whole sentence is difficult to read because of the simultaneous use of the words "that", "which" and "such".
We understand that this sentence was long and complex, making it difficult to read. We split the sentence to make it less complex and to avoid confusion, e.g. the issue raised in L49.

L49: Reads as "some of which" refer to eukaryotes.
This was changed in light of the comment r.e. L48-51.

L50: Why these biological roles are more important than others that are not mentioned here? Apparently the authors wanted to outline the effects that are of practical importance. Then this sentence should be revised accordingly.
It was not our intention to claim that these biological roles are more important than others. We chose to give some examples, therefore using 'such as', that are linked to the type of viruses we studied.

L51: "Among the RNA viruses, the double-stranded RNA (dsRNA) Totiviridae" Reads awkward. I suggest: "The double-stranded RNA viruses of the family Totiviridae"

Changed accordingly.

L55: delete "symbiotic" (see reasoning above)

Changed accordingly.

L56: it is > it has been

Changed accordingly.

L67: + Sauroleishmania

We thank the reviewer for this addition. It is true that LRV2 was recently identified in several isolates from Sauroleishmania species. We added the sub-genus name and a reference to the text.

L78: it also circulates in dogs, which are not wild. I suggest to delete this word

It is true that *L. braziliensis* has also been reported in dogs, but dogs have a very low potential as a reservoir for *L. braziliensis* (<http://dx.doi.org/10.1016/j.vetpar.2007.07.007>). As *L. braziliensis* is mainly linked to sylvatic life-cycles, in particular in Peru and Bolivia, it would be incorrect/incomplete to remove the word 'zoonotic'.

L81: Here, for example, mentioning species is already redundant.

Species name was removed as requested.

L82: another example: sampled Lb parasites > isolates

I will no more mention every instance of the studied species name and how to treat it.

Changed accordingly.

L86-90: Simplify the text: "We sequenced genomes of 79 isolates of *Leishmania braziliensis* with known LRV infection status, which had been sampled during various studies on the genetics and epidemiology of leishmaniasis 22,29–33. The read coverage ranged from 35x to 121x (median value - 58x)."

It is not needed to mention here the reference genome.

Changed accordingly.

L90-92: this sentence is fully redundant. It does not say anything about the data. Mentioning software here is unjustified.

We removed the sentence.

L92-94: It is unclear why these three isolates were removed. Please, bring back from the previous version the text explaining that.

We clarified the reason for excluding these three isolates and refer to the supplementary results for more information.

L95: unexplained abbreviation

MAF stands for Minor Allele Frequency. We thank the reviewer for noticing the absence of this explanation and added this accordingly in the text.

L99-100: What are these other eukaryotic genomes? The provided reference concerns Leishmania, I suggest mentioning just this genus. Moreover, I doubt that parasites these exact values are expected. Make a separate sentence like: "This is in line with what is known for the genomes of Leishmania spp."

Changed accordingly.

L105-106: It is unclear why only one curve from this graph is considered, while the three others show higher values. This must be properly explained.

We refer to both panels in the supplementary figure. In panel a, we show LD estimates for each of the four groups, showing that LD decreased to <0.2 in two groups. LD also decays strongly in the other two groups, but reaches plateau at ~0.3. In panel b we provide estimates of LD for the same groups but corrected for sample size. It is clear from this graph that LD decreases to <0.2 for all groups (and even <0.1 for two groups). We slightly altered the sentence to state these results more clearly.

L107: I am not sure, whether 0.2 is "around zero" in the scale from 0 to 1.

The reviewer is focusing on the two groups that have very low sample size. As shown in Supplementary Table 3, the standard deviation here is also large (0.4) and thus includes zero in the distribution. For the group with largest sample size (N=14), we estimated that FIS = -0.066 with a standard deviation of 0.245. This means that close to 70% of the FIS estimates for this group range between -0.311 and 0.179, and are thus around zero.

L112: differed at an average 9,866 fixed SNP differences > had in average 9,866 fixed SNP differences

This is not the exactly the same thing. We do mean that they differed on average by 9,866 fixed SNPs; our choice of words is thus correct.

L114: cluster > clustered

Changed accordingly.

L118: and one lineage > and one of them

Changed accordingly.

L123-124: "We identified three distinct ancestry components..." Specify the analysis that inferred this.

The type of analyses was added to the sentence.

L140: at > with

Changed accordingly.

L143: "three main parasite groups": You seemed to decide naming them as (ancestral) populations.

We changed 'main parasite groups' to 'ancestral populations'.

L146: Gray-scale > Gray scale

Changed accordingly.

L152: the earlier described > the identified (otherwise it reads as if these populations were described in a previous work. Moreover, the word described is not suitable here, since they are not much described so far.)

Changed accordingly.

L162-163: The differences consist in single nucleotide positions. Therefore these also could be random mutations, which occurred after the hybridization.

The number of SNPs in ADM is twice that of the number of SNPs observed in the ancestral populations. Given the slow mutation rates in eukaryote genomes, the most parsimonious explanation for such an extensive SNP diversity in the ADM hybrids is that they are the result of independent hybridization events. This was also further evidenced by the PCAdmix analyzes in the subsequent paragraph. We changed the sentence slightly to accommodate the doubts of the reviewer.

L166: to reference > to the reference

Changed accordingly.

L172: hybridization, not admixture, if we discuss event(s). Admixture is the result of such event(s).

Changed accordingly.

L179: Lb parasites > lineages

Changed accordingly.

L211-212: to either of the explanatory components > to individual components

Changed accordingly.

Figure 3: It would be more convenient if the values in the graph and in the text were in the same format, i.e. everything only as fractions (0.273) or percentage (27,3%).

We agree that this would be more convenient. We changed the fractions in Fig. 3 into percentages.

L241-244: The values in parentheses are distances between the sampling points, where these populations were documented. Geographic ranges are not the same as ranges of distances.

We agree with the reviewer and slightly altered the sentence to clarify that we are dealing with distances between sampling points rather than geographic ranges.

L255: analyzed > included; as revealed by previous > as revealed in a previous

Changed accordingly.

L268: Sequences > The assembled LRV1 sequences

Changed accordingly.

L280: LRV1 genomes of > LRV1 genomes from

Changed accordingly.

L281-283: Please, describe the topology of the tree. The position of LRV1 from *L. shawi*, breaking the clade of LRV1 from *L. guyanensis* must be mentioned. This apparently represents a rare random horizontal transition event, which nevertheless does not exclude the essential proportion of co-phylogeny between the other two *Leishmania* spp. and their viruses.

It is not the purpose of our study and our analyses do also not allow to confidently infer patterns of co-speciation and host-swithing (including the position of *L. shawi*) within the *L. guyanensis* species complex. This requires more in-depth analyses including additional genome data of the various *Leishmania Viannia* species.

L284-285: mention the two species just once (not thrice (!) in the sentence)

Changed accordingly.

L286-287: Dichotomy cannot be between something. Rephrase, or better unite this sentence with the next one, since actually both concern the same split.

The Cambridge dictionary states several examples of “dichotomy between” to describe a difference between two opposite ideas or things, for instance “This draws an obvious dichotomy between marine and freshwater systems”. Here we used it correctly to describe the dichotomy between L. braziliensis and L. guyanensis parasites.

Regarding the second comment of the reviewer, the two sentences are describing two different analyses. Supplementary figure 11 is a phylogenetic network of Leishmania genomes, while Figure 5B is a co-phylogenetic analysis with Leishmania and LRV1 genomes. While both show the deeper split for the Leishmania parasites, the former is presented to illustrate the ancestry of the parasite as was done for LRV1 in Figure 5A, while the latter is presented to show the co-ancestry of parasites and viruses.

L298: Co-divergence between > Co-divergence of
Changed accordingly.

L300-301: " LRV1 clades of Lb are coloured according to the different viral lineages identified in this study (L1-L9)" > LRV1 sequences from L. braziliensis are coloured according to the viral lineages identified in this study (L1-L9).

Changed accordingly.

Figure 5: please, mark the lineages L1-L9 on the tree to make it easier readable,
We marked the lineages’ names on the tree in panel A and agree that it increases readability.

L303: monophyletic LRV1 clades > LRV1 clades (clades are monophyletic by definition)

Changed accordingly.

L326: Viral sequences of the Bolivian Lb parasites > Viral sequences from the Bolivian isolates

Changed accordingly.

L329: the L7 viral lineage > the former

Changed accordingly.

L339-340: This is a repeat of the above information, delete the sentence.

Changed accordingly.

L353: " The distribution is shown for LRV+ Lb isolates belonging to the nine viral lineages" Lesishmania braziliensis isolates cannot belong to viral lineages. Suggested revision:

"The distribution is shown for the nine viral lineages"

Changed accordingly.

L362: phylogeny > dataset

Changed accordingly.

L374: Specify, that this is because of the low sample number for this group.

Changed accordingly.

L388: less > lower

Changed accordingly.

L415: dissemination > distribution; delete "endosymbiotic"

We believe 'distribution' is too narrow as a term here because our paper also includes calculations of viral dispersal rates. We thus believe that "dissemination" better captures the breadth of our work. 'Endosymbiotic' was deleted as suggested.

L416: Why these "here"? You cannot generalize these observations. Write directly about the objects of study.

We removed these to avoid repetition in the next sentence.

Then tons of unnecessary "Lb" in the discussion...

We see the point of the reviewer and altered the discussion in this regard. Hopefully there is now an appropriate amount of L. braziliensis throughout the discussion.

L448: "cycles of hybridization events". Why cycles? It is not a cyclical process. Just hybridization events.

Changed accordingly.

L455: was > has been

Changed accordingly.

L456: "in strains of L. (Viannia) species" > "for several species of the subgenus L. (Viannia)" OR "for several L. (Viannia) spp."

Changed accordingly.

L456: and also > as well as

Changed accordingly.

L458: clustered according to > clustered mostly according to (because this is not absolute)

Changed into "... mainly clustered according to ..."

L481: to the other viral lineages > to the others

Changed accordingly.

L482: that parasite gene flow > parasite expansion (?)

We understand this suggestion as we do show that the parasites are expanding from tropical rainforests. But expansion refers more to geographic ranges, while here we refer to the gene flow of the parasite.

L489-490: human migration, including movement of hemerophile reservoirs/hosts > migration of humans and hemerophile reservoir hosts

Changed accordingly.

L491, 505: endosymbiotic (see above)

Changed accordingly.

L493: "integrated into the Peruvian nation" Maybe "had poor contacts with the rest of the country"? Otherwise it reads absurd.

We really mean "integration in the Peruvian nation", because there was little or no exploitation of the Amazon region by the government. This all changed over the past century. Our sentence is thus correct.

L497: Amazon regions > the latter

Changed accordingly.

L501: replace comma with period and start a new sentence (from "This" instead of "which")

Changed accordingly.

L533: on a HOMEM medium > in the HOMEM medium; also specify the manufacturer

Changed accordingly.

L545: delete redundant "isolates"

Changed accordingly.

L550: > Were grown as above for 2-3 weeks. (avoid repeating the same information)

Changed accordingly.

L554: (NEB.) > (NEB)

Changed accordingly.

L558: " NovaSeq platform" – specify the model of the instrument;
generating an average of > generating in average

Changes were made as requested.

L587: p-distances between what?

We added the explanation of what p-distances are.

L596: with 50% as burn-in > with 50% burn-in

Changed accordingly.

L642 and elsewhere: IQtree > IQ-TREE

Changed accordingly.

L643: function > module

Changed accordingly.

L710: on > in

Changed accordingly.

L869-872: this is now a published paper

We thank the reviewer for his/her extensive feedback.

Reviewer #3 (Remarks to the Author):

Based on answers from authors to our queries, I found the manuscript acceptable as it is. They provide important information further supporting their study.

Not more to add.